# Ground-penetrating radar reveals ice thickness and undisturbed englacial layers at Kilimanjaro's Northern Ice Field

Pascal Bohleber[1,2,3], Leo Sold[4], Douglas R. Hardy[5], Margit Schwikowski[6], Patrick Klenk[1,10], Andrea Fischer[3], Pascal Sirguey[8], Nicolas J. Cullen[9], Mariusz Potocki[2,7], Helene Hoffmann[1], and Paul Mayewski[2]

[1]Institute of Environmental Physics, Heidelberg University, Heidelberg, Germany
[2]Climate Change Institute, University of Maine, Orono, ME, USA
[3]Institute for Interdisciplinary Mountain Research, Austrian Academy of Sciences, Innsbruck, Austria
[4]Department of Geosciences, University of Fribourg, Fribourg, Switzerland
[5]Climate System Research Center & Department of Geosciences, University of Massachusetts Amherst, Amherst, USA
[6]Paul Scherrer Institute, Villigen, Switzerland
[7]School of Earth and Climate Sciences, University of Maine, Orono, ME, USA
[8]National School of Surveying, University of Otago, New Zealand
[9]Department of Geography, University of Otago, New Zealand
[10]Now at: German Aerospace Center (DLR) Oberpfaffenhofen, Germany

*Correspondence to:* Pascal Bohleber (pascal.bohleber@iup.uni-heidelberg.de)

**Abstract.** Although its Holocene glacier history is still subject to debate, the ongoing iconic decline of Kilimanjaro's largest remaining ice body, the Northern Ice Field (NIF), has been documented extensively based on surface and photogrammetric measurements. The study presented here adds, for the first time, ground-penetrating radar (GPR) data at center frequencies of 100 and 200 MHz to investigate bed topography, ice thickness and internal stratigraphy at NIF. The direct comparison of the GPR signal to the visible glacier stratigraphy at NIF's vertical walls is used to validate ice thickness and reveals that the major internal reflections seen by GPR can be associated with dust layers. Internal reflections can be traced consistently within our 200 MHz profiles, indicating an uninterrupted, spatially coherent internal layering within NIF's central flat area. We show that, at least for the upper 30 m, it is possible to follow isochrone layers between two former NIF ice core drilling sites and a sampling site on NIF's vertical wall. As a result, these isochrone layers provide constraints for future attempts at linking age-depth information obtained from multiple locations at NIF. The GPR profiles reveal an ice thickness ranging between $(6.1 \pm 0.5)$ and $(53.5 \pm 1.0)$ m. Combining these data with a very high resolution digital elevation model we spatially extrapolate ice thickness and give an estimate of the total ice volume remaining at NIF's southern portion as $(12.0 \pm 0.3)10^6$ m$^3$.

## 1 Introduction

The ice masses on top of Kilimanjaro (Tanzania, East Africa), the "white roof of Africa", are the most recognized among the sparse glaciers in Africa. Three major ice bodies are found on the summit area of Kilimanjaro today (cf. entire mountain), Furwängler Glacier and the Northern and Eastern Ice Fields, which are remnants of a former ice cap which encircled the Kilimanjaro plateau at the end of the 19th century. Present-day climatological conditions are not favourable for maintaining

these glaciers and result in an overall negative mass balance of Kilimanjaro's glaciers (Hardy, 2002; Mölg and Hardy, 2004; Cullen et al., 2006, 2013; Mölg et al., 2008, 2009; Thompson et al., 2009; Hardy, 2011). The recent decline of Kilimanjaro's glaciers is well documented, with changes in glacier geometry derived from terrestrial and aerial photogrammetry as well as satellite imagery (Hastenrath and Greischar, 1997; Thompson et al., 2009; Cullen et al., 2006, 2013; Winkler et al., 2010; Sirguey and Cullen, 2014). Ground-based observations document ice loss by terrestrial laser scanning, comprehensive automatic weather stations (AWS) and network of mass balance stakes (Mölg and Hardy, 2004; Mölg et al., 2008; Hardy, 2011; Pepin et al., 2014); these data serve as input for modelling mass and energy balance (Mölg et al., 2003; Cullen et al., 2007; Mölg et al., 2009; Mölg and Kaser, 2011). In contrast to the extensive datasets from surface and aerial measurements, little is known so far about the underlying bed conditions and topography as well as ice thickness (Sirguey et al., 2013). Consequently, mapping ice thickness complements monitoring glacier decline and glaciological modelling of the past and future response of Kilimanjaro's glaciers to climate variability. This especially concerns the Northern Ice Field (NIF, Figure 1) because of two competing interpretations that exist regarding the maximum basal ice age and the mechanism of glacier formation. Ice cores have been drilled at several locations on NIF's central flat area (Thompson et al., 2002). The two ice cores that we refer to in the following, called NIF2 and NIF3, have been interpreted as continuous paleoclimate records, extending as far back as 11.7 ka BP (Thompson et al., 2002, 2009; Gabrielli et al., 2014). Based on observational as well as modelling considerations, Kaser et al. (2010) arrived at an alternative hypothesis, suggesting a cyclic build-up and decay of the tabular glaciers, with the ice likely coming and going repeatedly throughout the Holocene. New insights to resolve this ongoing controversy may come from utilizing NIF's vertical walls to sample directly the glaciers' stratigraphy for radiometric ice dating and ultra-high resolution sampling techniques. Previous attempts at radiocarbon dating of basal ice and also dust layers from vertical wall sampling has not yet definitively constrained NIF's glacier age (Thompson et al., 2002; Noell et al., 2014). Notably, only an undisturbed stratigraphy would allow a seamless link between results from wall samples and ice cores drilled in the interior of NIF. Complicating the situation is that the visible stratigraphy at some sections of the ice margin reveals inclined, converging layers. In addition, basal melting features attributed to isolated fumarole activities have been observed under plateau ice (Kaser et al., 2004), making stratigraphic disturbance by basal melting a possibility. It is thus not *a priori* evident to what degree stratigraphic integrity is preserved at NIF.

In this context, ground-penetrating radar (GPR) offers a powerful tool to investigate the geometry and internal structure of glaciers and ice sheets, making GPR nowadays a standard tool in glaciology (e.g., Dowdeswell and Evans, 2004; Navarro and Eisen, 2009). For non-polar glaciers, GPR is typically applied to study ice thickness, accumulation distribution and ice flow (Vincent et al., 1997; Binder et al., 2009; Campbell et al., 2012; Fischer and Kuhn, 2013). GPR has also been used successfully for mapping internal reflections in connection to ice cores on mountain glaciers (Pälli et al., 2002; Eisen et al., 2003; Konrad et al., 2013; Sold et al., 2015). On tropical glaciers, GPR has already been utilized successfully to determine ice thickness (e.g. Prinz et al., 2011; Salzmann et al., 2013; Chadwell et al., 2016). However, to our knowledge this is the first time a ground-penetrating radar survey was conducted at Kilimanjaro's Northern Ice Field. The NIF split in two separate ice bodies in 2012 (Cullen et al., 2013). We solely focus on the southern portion remaining on the summit (Drygalski and Great Penck) comprising the former ice core drilling sites. Hence we use the abbreviation "NIF" in the following to refer to this southern portion only.

Typical for the tabular glaciers on Kilimanjaro's summit (cf. slope glaciers) the NIF topography is characterized by a central flat plateau area and near-vertical ice margins (Kaser et al., 2004; Cullen et al., 2006; Hardy, 2011).

Our main objectives are to i) map bed topography and ice thickness, and ii) to study the internal stratigraphy of NIF through internal reflection horizons (IRH). In so doing, we devote special attention to evaluating the stratigraphic integrity of NIF be-

tween the ice core drilling area in NIF's interior and a sampling site on the vertical wall. Although not further discussed here, samples for radiometric age determination were obtained at this site on the vertical wall in a previous field campaign led by two of the authors (MS and DRH); results will be published elsewhere. Finally, we estimate the total ice volume presently remaining at NIF by spatially extrapolating the GPR-derived ice thickness.

## 2   Data and Method

The basic principle of a pulsed GPR system is to send an electromagnetic signal into the ground and to record the signal reflections as a function of their two-way travel time (TWT). Partial reflections of the electromagnetic wave recorded as IRH occur at vertical discontinuities in the dielectric material. From polar studies, IRH are known to coincide with variations in density, acidity (Robin et al., 1969), liquid water content (Forster et al., 2014) and changes in crystal orientation fabric (Fujita and Mae, 1994; Eisen et al., 2007). Only IRH connected to density and acidity variations are typically regarded as isochrones

(e.g., Navarro and Eisen, 2009). In view of its visible dust bands, potential presence of liquid water near surface and base and the absence of a firn column, it is not self-evident what physical causes of IRH can be expected to dominate at NIF.

### 2.1   Survey setup and data acquisition

GPR profiles were obtained over the course of three days during our expedition to the Kilimanjaro ice fields in September 2015. We used multiple GPR systems with center frequencies of 100, 200 and 800 MHz. Details of the technical settings and

data acquistion are summarized in Table 1. GPR profiles were obtained as common offset (CO-) profiles, i.e. transmitter and receiver are kept at a fixed distance while being moved over the glacier surface. Positioning was provided by conventional GPS receivers at approximately 5 m horizontal accuracy.

The spatial extent of the GPR survey was constrained by the tabular structure of NIF and by its rough surface terrain. The flexible 100 MHz antenna could be used over rough terrain found at large parts of NIF, especially close to NIF's northern margins.

The flat central area around the AWS and ice core drilling sites allowed us to also use sled-mounted systems. The 200 MHz sled antenna was used for mapping the spatial variation of the bed reflection as well as IRH within the flat central area. We also used an 800 MHz system for mapping shallow IRH (detected by 800 MHz within roughly the uppermost 10 m). Relative to the 200 MHz profiles, however, the 800 MHz profiles did not provide additional information and are not further discussed here. An overview on the spatial coverage provided by all CO-profiles is presented in Figure 1. The glacier outline shown in

Figure 1 was digitized based on an ortho-rectified GeoEye-1 satellite image acquired on 23 October 2012 (Sirguey and Cullen, 2014), consistent with the methodology described in Cullen et al. (2013). Since the corresponding satellite image was recorded in October 2012, this procedure includes a minor overestimation of the ice margins (estimated in section 2.5 below), which

have been continuously retreating.

Using an additional 200 MHz system with separable receiver and transmitter, a common-midpoint profile (CMP) was performed at a central location within the drilling area (Figure 1). Due to technical difficulties in the field, only a maximum antenna separation of 7 m could be achieved and only a single CMP was recorded. With one-sided distances: $0.1, 0.3, ..., 3.5$ m, a number of N$=$ 18 shots were obtained starting from center, symmetric and synchronous.

## 2.2 Post-processing of GPR data

We used the standard routines to process GPR data (using Reflexw, Sandmeier Geophysical Research) including static correction, bandpass-filtering and adding a gain (to compensate for geometrical divergence losses). As opposed to the wheel-triggered 200 MHz CO-measurements, the 100 MHz measurements were acquired by time-triggering. Thus, they were interpolated to equidistant shots (0.25 m) based on co-registered GPS data. We employed Kirchhoff migration using a summation width of 5 traces. Because of the insignificant amount of firn at NIF, we used the pure-ice value of v$=$ 0.168 m/ns for the electromagnetic wave speed (e.g., Navarro and Eisen, 2009). The same constant wave speed of v$=$ 0.168 m/ns was used for traveltime-depth conversion. Different settings were tested and this processing scheme provided the best results regarding visibility of internal and bed reflection. An illustration of the processed GPR data for a direct comparison of 100 and 200 MHz CO-profiles is shown in Figure 2. In the 200 MHz CO-profiles, IRH were traced visually and additionally supported with a semi-automated phase-following routine. The bed reflection horizon was tracked visually. For the CMP, a semi-automated feature tracking routine was used to pick the signal of two internal reflections (Figure 3). A hyperbolic fit to these picked reflections yielded an estimated near-surface permittivity of $3.22 \pm 0.17$ and $3.21 \pm 0.35$, for the uppermost $(3.15 \pm 0.11)$ m and $(4.62 \pm 0.30)$ m depth, respectively. This corresponds to a wave speed of $(0.167 \pm 0.004)$ m/ns and $(0.167 \pm 0.009)$ m/ns, respectively.

## 2.3 Uncertainty considerations

Major contributions to uncertainty in depth of an IRH come from i) the vertical resolution provided to determine the TWT of the IRH and ii) uncertainty in the electromagnetic wave speed, in our case especially related to the presence of near-surface meltwater.

Contribution i) depends on the extension of the GPR pulse and is typically assumed as half the wave period, or 5 and 2.5 ns for 100 and 200 MHz, respectively (Navarro and Eisen, 2009). For picking an IRH (200 MHz) an additional uncertainty component stems from potentially losing track of the individual coherent phase used for tracing the reflection. Based on the typical separation in phases of the IRH the related error was estimated as 4 ns, and twice as much in case of the bed reflection. The combined uncertainty of the picked travel times is then 5 and 8 ns for IRH and bed reflection at 200 MHz, respectively and 9 ns for bed reflection at 100 MHz (calculated by error propagation, root sum of squares). Regarding contribution ii), wave speed values derived from the shallow CMP are within $1\%$ of the pure-ice value 0.168 m/ns. The negligible amount of firn reported in the NIF ice cores (Thompson et al., 2002) suggests that it is a valid assumption to neglect firn and snow layers. However, a spatially variable amount of percolating meltwater (visible in the CO-profiles and further discussed in section 3 below) implies locally increased material permittivity and hence lower wave speed values. Additional quantitative information

on water content within the ice column would be needed for a precise calculation of wave speed variability due to meltwater. At the position of the CMP, the CO-profiles do not show exceptionally strong meltwater presence (corresponding to point "intersection" in Figure 4). Accordingly, the difference of $1\%$ between the CMP estimate and the wave speed of pure ice is regarded as an adequate uncertainty for sections without meltwater and only as a lower uncertainty limit where meltwater is

present. Based on these considerations, typical uncertainties are around 1 m for detecting an IRH and around 1–2 m for the bed reflection (and ice thickness). In addition, in case of a strong surface/bed inclination the accuracy of the GPR ice thickness can be limited to less than $16\%$ if only a 2D migration is performed (Moran et al., 2000). A full 3D migration based on a dense survey setup was beyond the scope of this work, given that a mostly planar bed is expected at NIF.

Shot distances in data acquisition were chosen less than one quarter wavelength apart in order to avoid spatial aliasing (Table 1).

This also holds for the 100 MHz measurements that were recorded at a constant time interval of 0.5 s while pulling the antenna at a walking speed of about 0.5 m/s. Horizontal resolution of the properly migrated radargrams can thus be estimated as half of the wavelength, independent of reflector depth (Welch et al., 1998; Yilmaz, 1987). This corresponds to approximately 80 cm and 42 cm of horizontal resolution for the 100 MHz and 200 MHz profiles, respectively.

## 2.4   Validation of traveltime-depth conversion at vertical ice cliff

We used the vertical wall at the southern margin to directly compare the ice thickness derived from GPR with a photogrammetric estimate. The 200 MHz CO-profile running towards the vertical wall (Profile 1 in Figure 4) ends within about one meter from the cliff and shows an ice thickness at the cliff of $(37.0 \pm 1.5)$ m (Figure 4). The height of the ice cliff was independently estimated by hanging a 16 m long rope with a weight at the end from the edge. Using the known length of the rope as a reference in a deskewed picture of the cliff, obtained by photogrammetric processing of 17 multi-view oblique photographs

(using Agisoft Photoscan), yields an independent estimate of the total height of the cliff of 38 m. To derive a lower estimate of uncertainty, we assumed 0.3 m uncertainty in the length of the rope at 16 m (resulting from knots tied into the rope) and neglected streching of the rope. This translates to $(38.0 \pm 0.7)$ m. Further uncertainty is introduced by the image stitching and deskewing routines. The software estimates the internal and external camera orientation from the image data alone. Hence, the quality of the results strongly depends on the camera positions, image overlap and the object shape (Agisoft, 2016). In

comparable applications, related errors in the millimeter and low centimeter range were found (e.g Thoeni et al., 2014; Prieto and Ramos, 2015). In our case they cannot be quantified and were assumed to be negligible.

## 2.5   Interpolation of ice thickness

To derive the ice thickness distribution over the NIF from our 100 and 200 MHz profiles, we essentially followed the approach previously developed by Fischer (2009), first interpolating the bed topography and then computing the difference between

surface and bed elevation. This method (here referred to as "Grid") allowed us to use not only the GPR ice thickness measurements for the spatial interpolation, but also additional topographic information from the existing digital elevation model (DEM) KILISoSDEM2012 and the position of the glacier margins (Sirguey and Cullen, 2014). The DEM provides high resolution (0.5 m x 0.5 m) data of the 2012 surface at Kilimanjaro summit area with 2.12 m LE90 ($90\%$ percentile linear error)

accuracy. No densely distributed information is available regarding changes in surface altitude over the entire NIF surface between the acquisition of the DEM (2012) and our radar survey (2015). At the NIF central flat area, however, ablation stake measurements show almost no systematic change in mean surface elevation between 2012 and 2015 (Figure 5). We did not use the vertical coordinate of our conventional GPS measurements for estimating ice thickness, since no differential GPS was used

and thus the uncertainty in altitude is likely larger than the expected altitude change between 2012 and 2015.

The contour lines of bed elevation below NIF were drawn in 20 m equidistance constrained by point values calculated by subtracting the GPR measured ice thickness from the surface elevation (DEM). This was done at around 100 different positions thus using only a minor subset of all GPR datapoints. Next, the subsurface bed topography was interpolated within the glacier outline using the Topo2raster tool in the ArcGis 10.2 software (Hutchinson et al., 2011) (Parameters: no enforcing, 5 m

gridsize). Based on the interpolated bed topography, distributed ice thickness was calculated as the difference between surface DEM and interpolated bed.

We derived an estimate of the total ice volume by multiplying the mean ice thickness by the total surface area. For estimating the uncertainty in mean ice thickness, we used the GPR datapoints that were previously not used for interpolation. Calculated at the positions of the respective GPR datapoints, the mean of the difference in ice thickness between GPR and interpolation is

$(0.63 \pm 2.24)$ m (small insert in Figure 7). To estimate the surface area lost between October 2012 (satellite image) and September 2015 (our expedition), the rate of area change reported for NIF by Cullen et al. (2013) for the time period 2011.46–2003.08 (Table 2 in (Cullen et al., 2013)) was used. With an annual surface area loss of $-0.447 \ 10^{-2} \ \mathrm{km}^2\mathrm{yr}^{-1}$ this leads to a correction of the surface area from $0.525680 \ \mathrm{km}^2$ (October 2012, from satellite image) to $0.512643 \ \mathrm{km}^2$ (September 2015). The latter value was used to calculate the 2015 ice volume (Table 2).

For comparison with our combined GPR-DEM approach, we also considered interpolation based solely on the DEM and GPR, respectively. For the latter, we applied ordinary kriging directly to the GPR ice thickness profiles. To allow the interpolation of the sparse data, a large grid size had to be chosen (100 m) and the ice thickness at the outline vertices was set to zero. Although clearly suffering from these restrictions, we included the use of kriging for comparison as a method based on the GPR datasets only. In an approach similar to the "Grid" method, interpolation based on the DEM only was done by removing all data points

over the NIF (including a 10 m buffer) and interpolating the void using Topo2raster (Sirguey et al., 2013) (in this case without additional constraints from GPR, however). Notably, the DEM-based approach includes no data from 2015, thus resembling conditions in October 2012.

## 3 Results and Discussion

Figure 2 shows processed radargrams from two parallel sections of 100 and 200 MHz CO-profiles. Both profiles display a

clear reflection of the underlying bed (e.g. at TWT of 550–650 ns in Figure 2) and some near-surface signal disturbance due to meltwater (e.g., between 80 and 100 m along the profile in Figure 2). Coherent internal reflectors are well represented in all 200 MHz profiles (included as Supplementary Material), but appear only to a limited extent in 100 MHz profiles (due to the coarser vertical resolution at lower frequency). The following discussion of results focuses on three main features of the radar

profiles, namely: i) bed reflection and ice thickness estimation, ii) internal layer architecture within the NIF central flat area, and iii) meltwater disturbance.

## 3.1 Mapping ice thickness at NIF

The validation of the GPR-derived ice thickness at the ice cliff with a photogrammetric estimate confirms a reliable estimation of ice thickness when using the constant pure-ice wave speed for the traveltime-depth conversion. For intersecting or overlapping 100 and 200 MHz profiles, the TWT of the bed reflection and hence also values for ice thickness are consistent within their uncertainty, typically within less than 1 m (Figure 2 and Figure 4).

Figure 6 shows the color-coded ice thickness along all acquired common-offset GPR profiles. Ice thickness ranges from around $(6.1 \pm 0.5)$ m at the western margin to a maximum ice thickness of $(53.5 \pm 1.0)$ m on the eastern part of the central flat area. Ice thickness within the central drilling area is typically around 46 m. At the ice core drilling sites NIF2 and NIF3, ice thicknesses of 50.8 m and 49 m respectively, were reported by Thompson et al. (2002) for the year 2000, without uncertainty. In 2015, our GPR derived ice thickness at NIF2 and NIF3 is $(44.7 \pm 1.7)$ m and $(42.4 \pm 1.5)$ m, respectively. This corresponds to a loss in ice thickness of $(6.1 \pm 1.7)$ m and $(6.6 \pm 1.5)$ m at NIF2 and NIF3, respectively, between 2000–2015. Since neither NIF2 nor NIF3 feature large surface/bed inclination (migration issues) nor pronounced presence of meltwater (Figure 4) the uncertainty in GPR ice thickness seems to be well represented by our previous considerations. For the time period 2000–2015, ablation stakes at the NIF plateau show an average change in surface elevation of around $-4.0$ m, with an uncertainty range between $-3.4$ and $-5.3$ m (Figure 5, bottom plot). For February 2000 to 15 September 2015, the cumulative surface height change measured by two ultrasonic sensors at the AWS, close to NIF2, is $-4.24$ m.

Although ice loss values obtained from the GPR-ice core comparison and ablation stakes agree within their estimated uncertainties, it seems worth mentioning that GPR-ice core derived ice loss is systematically larger than the ablation stake measurements. In this context we also note that, in principle, a contribution to the difference between ice thickness and surface elevation change could result from basal melting. Basal melting caused by fumarole activity has been observed at NIF (Kaser et al., 2004) and by two of the authors (DRH and NJC) at multiple locations. On the other hand, our GPR data generally shows no clear evidence of basal cavities that could result from pronounced subglacial fumarole activity. To match the observed difference between ice thickness and surface elevation change, basal melting would need to occur below the central flat drilling area on NIF, at a slow rate and without large spatial gradients. In the absence of GPR evidence for basal fumarole activity and lacking quantitative information on basal melting, it seems more likely to attribute the observed systematic difference in the two ice loss estimates to the uncertainties involved in GPR and ablation stake measurements, combined with spatial variability of ablation rate and, to a minor extent, a potential discrepancy in the ice core length.

The interpolated ice thickness distribution is shown in Figure 7. Ice at NIF reaches a maximum thickness of 54.0 m at approximately the highest-elevation area of the glacier, along an east-west trending ridge roughly at the center of the remaining ice field. A second area exceeding 40 m in thickness was identified towards the eastern end. A large area of ice thicknesses less than 10 m is found towards the western margin. The low ice thickness is likely a result of the surface gradually sloping off towards the west outside the caldera. A distinct rise in the local GPR bed reflection appears where the location of the crater rim

below the ice is suggested by satellite images (Figure 6, b)). Accordingly, the assumption of a generally flat bed topography does not hold everywhere below NIF. This finding supports the idea that local bed relief features may have affected past ice build up and decay through limiting exposure to solar radiation and wind (Kaser et al., 2010).

Table 2 summarizes our estimates of mean ice thickness and ice volume based on combining GPR and DEM ("Grid") and the interpolation based on GPR (Kriging) and DEM only, respectively. The assumption of zero ice thickness at the margins used for kriging certainly does not apply to the western margin. This results in an underestimation of volume as compared to the Grid approach. In addition considering the coarse resolution used in the kriging approach, we interpret the ice thickness derived from this method with caution only. The estimates of total ice volume obtained from the Grid approach and DEM-only are $(12.0 \pm 0.3)$ and $(14.3 \pm 1.3)$ $10^6$ m$^3$, respectively. The main contribution to the difference in ice volume comes from different mean ice thickness values as opposed to surface area (using the 2012 surface area the mean ice thickness obtained from the Grid method gives a volume of $(12.3 \pm 0.3)$ $10^6$ m$^3$). The decrease in mean ice thickness suggested by the comparison of the two interpolation methods is not supported by surface height change measurements 2012–2015. Since both interpolation methods use the same surface topography supplied by the DEM as input, the difference in mean ice thickness has to come from differences in determining subglacial bed topography. Consequently, the difference in ice volume estimates is not used to infer a rate of ice loss. Integrating both the DEM and GPR as constraints, the Grid method provides the most reliable ice volume estimate. We acknowledge that i) one volume estimate does not allow us to infer retreat rates and ii) for predicting the expected lifetime of NIF under ongoing ice loss conditions, a simple linear extrapolation based on current rates of lateral and surface retreat likely produces unrealistic values. Nonetheless, this first-ever quantification of NIF's ice volume based on direct GPR measurements of ice thickness provides an important context to the discussion of ongoing glacier retreat (Thompson et al., 2009; Mölg et al., 2010; Thompson et al., 2010).

## 3.2 Internal layer architecture within the NIF plateau

All 200 MHz profiles contain a number of coherent internal reflection horizons except for the lowermost depths. Below typically about 30 m, reflections still appear in intervals but cannot be traced continuously. Sections lacking echos from deep layers often coincide with a large amount of near-surface scattering, presumably due to the presence of near-surface meltwater. Absorption from meltwater causes less energy to be returned from deeper layers and hampers the detection of deep IRH. This explanation implies that coherent internal layers may still exist at greater depths but cannot be detected continuously anymore by GPR. It is worth noting that the vertical cliffs show instances of tilted and converging layers in close proximity to bed (Figure 8) which can also hamper the detection of deep reflectors. We believe that this stratigraphic convergence is an ablation feature rather than due to rheology (e.g. localized shearing at the glacier margin), as localized shearing appears evident only near the snout of the steepest slope glaciers, and features such as that shown in Figure 8 occur elsewhere on Kilimanjaro glaciers, particularly on the south side. The GPR profiles towards the western end are the only case in which adjacent IRH (representing boundaries to a layer of ice) are found merging together (see Supplementary Figure 1, Profile D). While we find no evidence of converging IRH in the central flat area of NIF, it is not possible to generalize this result also for the lowermost meters of basal ice where distinct IRH are absent.

The comparison of the GPR signal with the photo of ice cliff shows that distinct reflectors occur at depths where dark dust bands are visible (Figure 4). The glacio-chemical analysis performed by Thompson et al. (2002) on the NIF ice cores shows that dust layers coincide with a strong increase in the concentrations of nearly all ion species, including ammonium and chloride. It is plausible that the according change in the electrical conductivity of the ice layer produces a strong reflector seen in

the GPR data (Sold et al., 2015). Accordingly, this strongly suggests dust layers being a main physical cause of IRH at NIF. Thompson et al. (2002) and Gabrielli et al. (2014) report visible dust layers in the NIF2 and NIF3 ice cores (1 layer at 32.5 m in NIF3, 4 layers at 26, 29 m, and two each around 32 m in NIF2, all in reference to 2000). With the upper ice surface thinning estimated (from difference between the borehole depth (2000) and the GPR-derived ice thickness (2015)) as $(6.1 \pm 1.7)$ and $(6.6 \pm 1.5)$ m for NIF2 and NIF3, respectively, these layers are expected at around $(25.9 \pm 1.5)$ m (NIF3) and $(19.9 \pm 1.7)$,

$(22.9 \pm 1.7)$ m and $(25.9 \pm 1.7)$ m (NIF2) depth in 2015. We visually identified five prominent reflectors, consecutively labeled IRH 1–5 with increasing depth (Table 3), in profile 1 shown in Figure 4.

In order to trace IRH 1–5 along multiple profiles, they are linked at the intersections of the profiles by checking for consistent TWT, or depths (Figure 4 and Table 3). Connecting all 200 MHz profiles in this manner, IRH were followed proceeding along a closed course, successfully demonstrating that it is possible to return to the same depth/traveltime after a complete round

course. While it is possible to trace IRH 5 between the vertical wall and the drilling sites NIF2 and NIF3, IRH 4 is the deepest reflector that is traceable almost uninterruptedly throughout all profiles, with a short exception towards the eastern end and below the crater rim towards the west (Figure 9). This spatial extension of IRH within NIF suggests that, at least within the area mapped by 200 MHz profiles, IRH stem from continuous reflecting surfaces that can be associated with a corresponding dust layer. We thus conclude that the internal stratigraphy within the NIF central flat area is generally composed of uninterrupted,

spatially coherent layers (as opposed to deformed, macroscopically disturbed layers). A potential exception to this finding is ice just above the bed where GPR can neither support the existence of disturbances nor their absence.

The continuous layering mapped by GPR demonstrates that, in general, the internal layering is intact between the ice margin and the interior of the NIF plateau area. More specifically, the link between IRH and major dust layers implies that IRH represent isochrones and, thus can be used to extrapolate and compare age-depth information. This GPR-based tracking of

isochrones has been employed successfully not only on polar ice sheets but also at small scale mountain drilling sites (Eisen et al., 2003; Konrad et al., 2013). At NIF, the tracing of IRH provides a quantitative link between isochrone depths at existing sampling sites, thereby revealing important constraints for future efforts at integrating age-depth information obtained from the NIF ice cores and the vertical wall. Table 3 summarizes the respective isochrone depths obtained from tracing IRH 1–5 between NIF2, NIF3 and the vertical wall sampling site (cf. Figure 4).

With respect to the two ice core drilling sites, related isochrone layers are found at lower relative depth at NIF3 than at NIF2 (Table 3). Comparing the main features of the stable water isotope records of the NIF2 and NIF3 ice cores, Thompson et al. (2002) developed a matching of the two ice core depth scales that is qualitatively going in the same direction (i.e. Figure 4 D and Table 3 of this study in comparison with Figure 2 in Thompson et al. (2002)). On a quantitative level, however, tracing IRH between NIF3 and NIF2 yields tie points that are systematically at lesser depth in NIF2 as compared to the ice core stable

isotope matching. For instance, the thick layer at $(25.9 \pm 1.7)$ m (for 2015) in NIF3 (reported as 30 mm thick by Thompson

et al. (2002)) appears to correspond with IRH 5 found at NIF3 around $(26.6 \pm 0.6)$ m. Thompson et al. (2002) interpreted this layer in NIF3 to be aligned with the base of NIF2 (17.5 m deeper). Our findings raise questions about this interpretation (Table 3). In this respect it is worth noting that our findings do not change significantly if the average change in surface elevation of around $-4.0$ m is used in the above correction for the 2000–2015 surface thinning.

## 3.3 Effects of near-surface meltwater

As illustrated by the 100 and 200 MHz profiles in Figure 2, incoherent near-surface noise in 200 MHz radargrams coincides with increased near-surface reflectivity in the 100 MHz data. This characteristic is observed throughout all profiles at great spatial variability, and is interpreted as backscatter due to meltwater. This effect can extend to substantial depths (at times more than 10 m), probably where meltwater percolates through cracks or small crevasses. The abundant presence of englacial meltwater was confirmed by shallow mechanical drillings at the NIF central area during the 2015 expedition, and has been observed intermittently since February 2000 by one of the authors (DRH). Even during the early morning hours, shallow boreholes (around 0.6 m depth) filled with meltwater in 15–20 minutes. Hence our GPR profiles demonstrate a highly heterogeneous presence of meltwater near the surface, apparently a wide-spread feature at NIF related to spatial and temporal variability in surface characteristics and processes (Hardy, 2011). This finding is of relevance for any new ice core drilling efforts at NIF in the future, suggesting that chemical and isotopic records of the upper 10 m or more could be potentially corrupted by meltwater. The wide-spread presence of near-surface meltwater also needs to be considered in future energy and mass balance modelling efforts (Mölg and Hardy, 2004). Further quantifying the generation and evolution of the near-surface meltwater distribution points to important future research questions at NIF.

## 4 Conclusions and Outlook

The application of ground-penetrating radar at Kilimanjaro's Northern Ice Field provided a direct estimate of the remaining ice volume. For the central former drilling area, the radar profiles reveal macroscopic coherent, uninterrupted ice layering for at least the upper 30 m, and demonstrate abundant melt water in the top 10 m. The latter finding suggests that the upper part of future chemical and isotopic ice core records could potentially be corrupted by meltwater. The association of internal reflections seen by GPR with dust layers becomes evident from using NIF's vertical walls to compare the local GPR signal to the visual stratigraphy. The internal reflections were traced consistently within our 200 MHz profiles, indicating that the stratigraphic integrity is preserved. Tracing internal reflections provided a link of isochrone depths among the former ice core drilling sites and the vertical wall sampling site. This link implies valuable constraints for future efforts at integrating age-depth information obtained from the NIF ice cores and the vertical wall. Accordingly, our results contribute to future attempts at resolving the ongoing debate on NIF's age structure and glacier history.

For the first time on NIF, our GPR measurements provided widespread ice thickness soundings. In combination with the existing digital elevation model this allowed us to estimate the total ice volume remaining at NIF's southern portion as $(12.0 \pm 0.3) \, 10^6$ m$^3$. These data contribute to the understanding of ongoing glacier loss and will support existing glacier monitoring

databases. Regarding future drilling efforts at NIF, the presented data can aid the selection of potential coring sites through the newly gained information on ice thickness and bed topography, but also the heterogeneity in the presence of liquid water near the surface. Although connected to substantial logistical effort, repeat measurements of ice thickness would offer a precise method to support future studies on the ice loss at NIF, especially in terms of spatial variability. Moreover, the application of

5 GPR could be extended with great benefit also to monitor ice thickness at the other major ice bodies remaining on Kilimanjaro.

**Data availability**

Ice thickness along all radar profiles are available at https://doi.org/10.1594/PANGAEA.867908

*Acknowledgements.* We gratefully acknowledge financial support by National Geographic Society Science and Exploration Europe, grant GEFNE123–14. Additional financial support to PB was provided by the Deutsche Forschungsgemeinschaft (BO4246/1–1), and DRH has

10 been supported by the US National Science Foundation (NSF), and NOAA Office of Global Programs, Climate Change Data and Detection Program (0402557 and NSF ATM-9909201). We greatly acknowledge the generous offer of Prof. Kurt Roth (Heidelberg University) and his group to lend us their 200 MHz GPR system. We thank Chiara Uglietti (Paul Scherrer Institute) for her helpful comments. Simon Mtuy (SENE Mbahe, Tanzania) and his team of guides and porters are gratefully acknowledged for their logistical support during the expedition. We also would like to thank Denis Samyn, an anonymous reviewer and editor Kenny Matsuoka for their thorough reviews and helpful

15 suggestions.

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

**Table 1.** Overview of GPR systems used and data accquisition settings. In case of the CMP, number of samples refers to the number of shots to obtain the profile.

| Frequency (MHz) | Manufacturer | Platform | Shielded (Yes/No) | Profile | Trigger | Shot distance | Time window (ns) | Samples |
|---|---|---|---|---|---|---|---|---|
| 100 | Mala Geoscience | Rough-terrain | No | CO | Time | 0.5 s | 996 | 2024 |
| 200 | Ingegneria Dei Sistemi | Sled | Yes | CO | Wheel | 0.05 m | 650 / 800 | 8192 |
| 200 | Mala Geoscience | Separable | No | CMP | Manual | 0.1, 0.3, ..., 3.5 m | 100 | 18 |
| 800 | Mala Geoscience | Sled | Yes | CO | Time | 0.5 s | 138 / 277 | 2024 |

**Table 2.** Ice thickness derived from spatial interpolation using a combination of GPR and DEM data ("Grid"), GPR only ("Kriging") and DEM only ("DEM"). Uncertainties for Grid and Kriging are estimated from comparison with unused GPR datapoints (see text). For the DEM method uncertainties are reported as one standard error.

| Method | | Grid | Kriging | DEM |
|---|---|---|---|---|
| Range | (m) | $2.0 - 54.0$ | $0.0 - 53.5$ | $0.0 - 55.5$ |
| Mean thickness | (m) | $23.3 \pm 0.6$ | $21.2 \pm 1.0$ | $27.2 \pm 2.5$ |
| Ice volume | ($10^6$ m$^3$) | $12.0 \pm 0.3$ | $10.9 \pm 0.5$ | $14 \pm 1$ |
| Area | (m$^2$) | 512643 | 512643 | 525680 |
| Date | | 09/2015 | 09/2015 | 10/2012 |

**Table 3.** : Two-way travel times and (absolute and relative) depths of internal reflections (IRH) traced between the ice core drilling sites NIF2 and NIF3 and the vertical wall sampling site ("wall"). Horizontal distances are measured along the profiles from their intersection. Profile numbers refer to the legend in Figure 4.

| Profile | 1 | 1 | 1 | 2 | 2 |
|---|---|---|---|---|---|
| Position | wall | NIF3 | intersection | intersection | NIF2 |
| Distance [m] | 35 | 60 | 0 | 0 | 45 |
| IRH 1 [ns / m / %] | 74 / 6.2 / 17 | 84 / 7.1 / 17 | 86 / 7.2 / 17 | 87 / 7.3 / 17 | 91 / 7.6 / 17 |
| IRH 2 [ns / m / %] | 108 / 9.1 / 25 | 118 / 9.9 / 23 | 119 / 10.0 / 23 | 121 / 10.2 / 24 | 131 / 11.0 / 25 |
| IRH 3 [ns / m / %] | 187 / 15.7 / 42 | 198 / 16.6 / 39 | 203 / 17.1 / 40 | 202 / 17.0 / 40 | 221 / 18.6 / 42 |
| IRH 4 [ns / m / %] | 266 / 22.3 / 60 | 270 / 22.7 / 54 | 280 / 23.5 / 55 | 283 / 23.8 / 56 | 325 / 27.3 / 61 |
| IRH 5 [ns / m / %] | 324 / 27.2 / 74 | 317 / 26.6 / 63 | 331 / 27.8 / 65 | 333 / 28.0 / 66 | 396 /33.3 / 74 |
| bed [ns / m / %] | 440 / 37.0 / 100 | 505 / 42.4 / 100 | 507 / 42.6 / 100 | 503 / 42.3 / 100 | 532 / 44.7 / 100 |

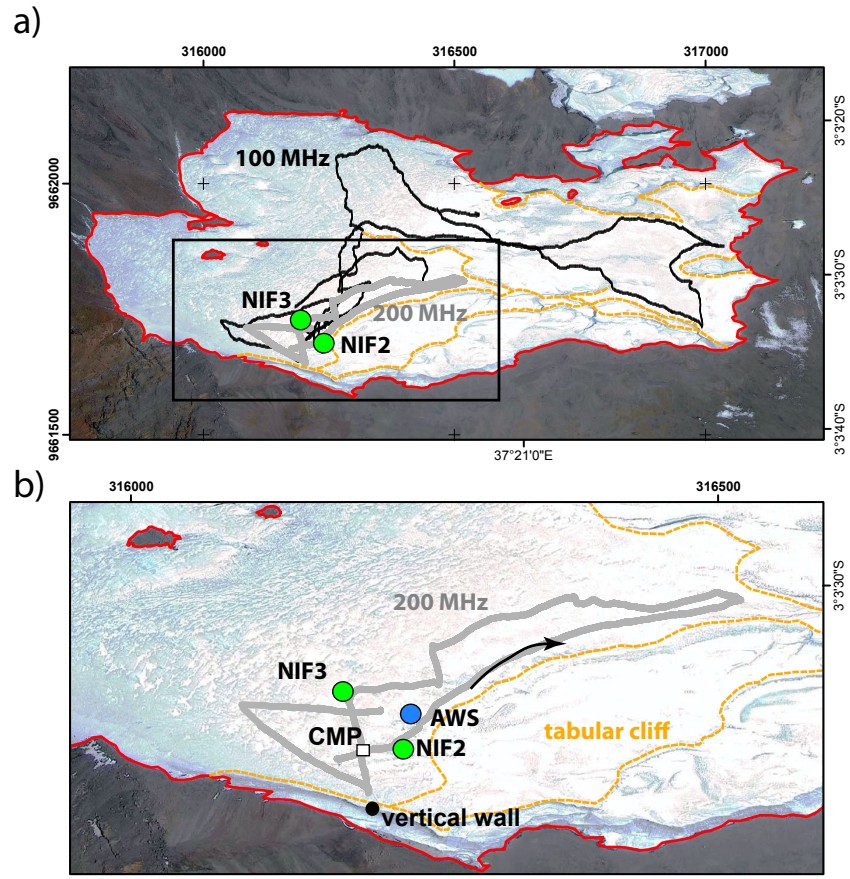

**Figure 1.** Spatial coverage of GPR common-offset profiles. Showing in a) are all 100 and 200 MHz profiles in black and grey lines, respectively. The zoom-in window b) (corresponding to the black rectangle in a)) shows the 200 MHz profiles only. The black arrow indicates the approximate position of the two profiles compared in Figure 2. Orange dashed lines highlight the tabular cliffs of NIF. Coordinates are in UTM 37M.

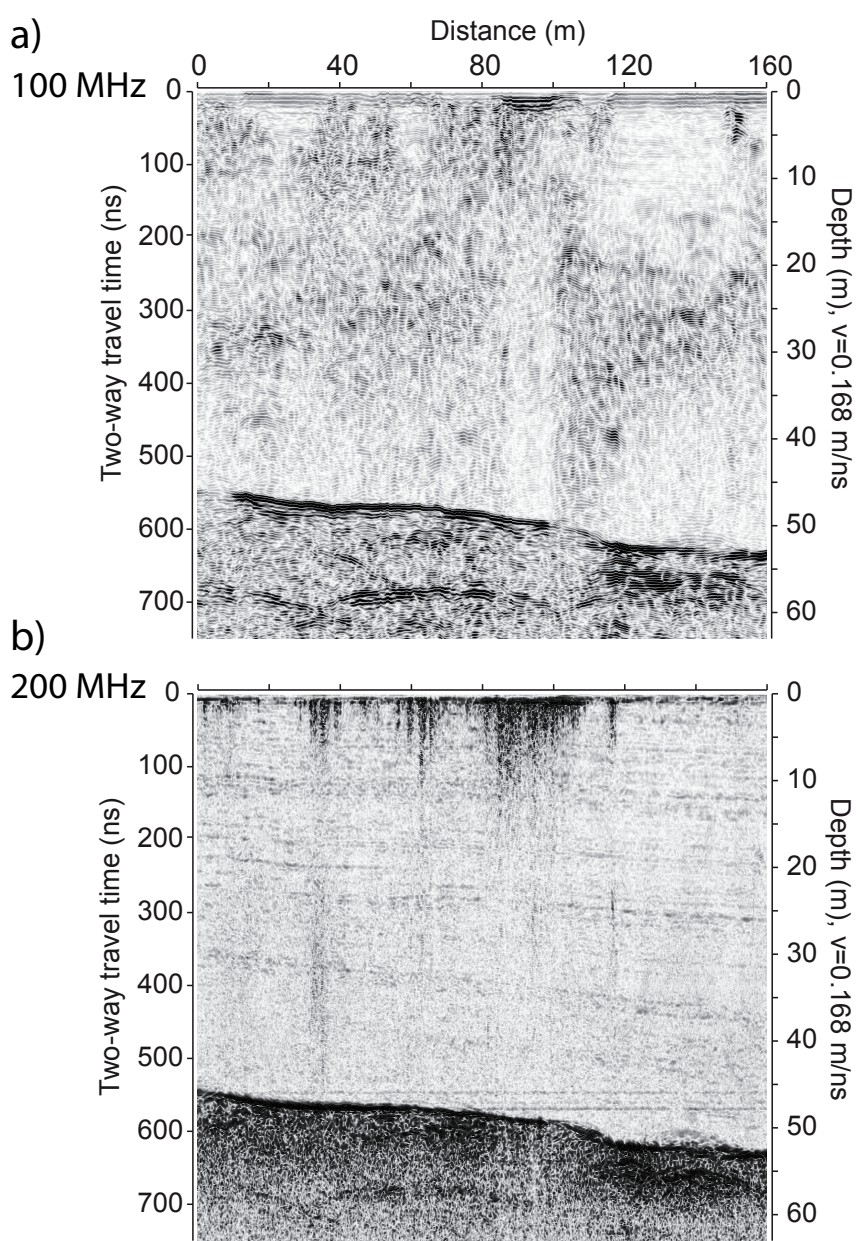

**Figure 2.** Example of processed radargrams recorded with 100 (a) and 200 MHz (b) over the same profile (position shown in Figure 1). Increased near surface reflectivity at 100 MHz coincides with incoherent noise near surface at 200 MHz (esp. between 80 and 100 m) and is interpreted as meltwater. Note that the horizontal lines in 200 MHz around 550 and 570 ns are artifacts.

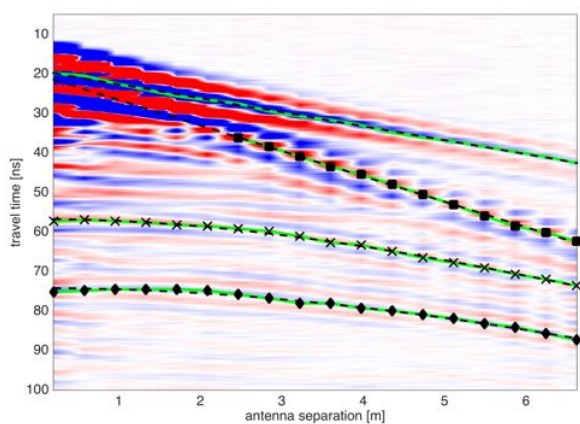

**Figure 3.** Evaluation of common midpoint profile. Picking was executed by a semi-automated algorithm tracing a center feature of the respective pulse throughout the radargram. Green lines show picks for direct air and ground wave (no symbol and dots, respectively) and internal reflections (crosses and diamonds). The lower two reflectors (crosses, diamonds) were used for permittivity and wave speed estimation. For this purpose a hyperbola fit (dashed line) was calculated based on the picked values using the time zero offset as derived from the direct air wave.

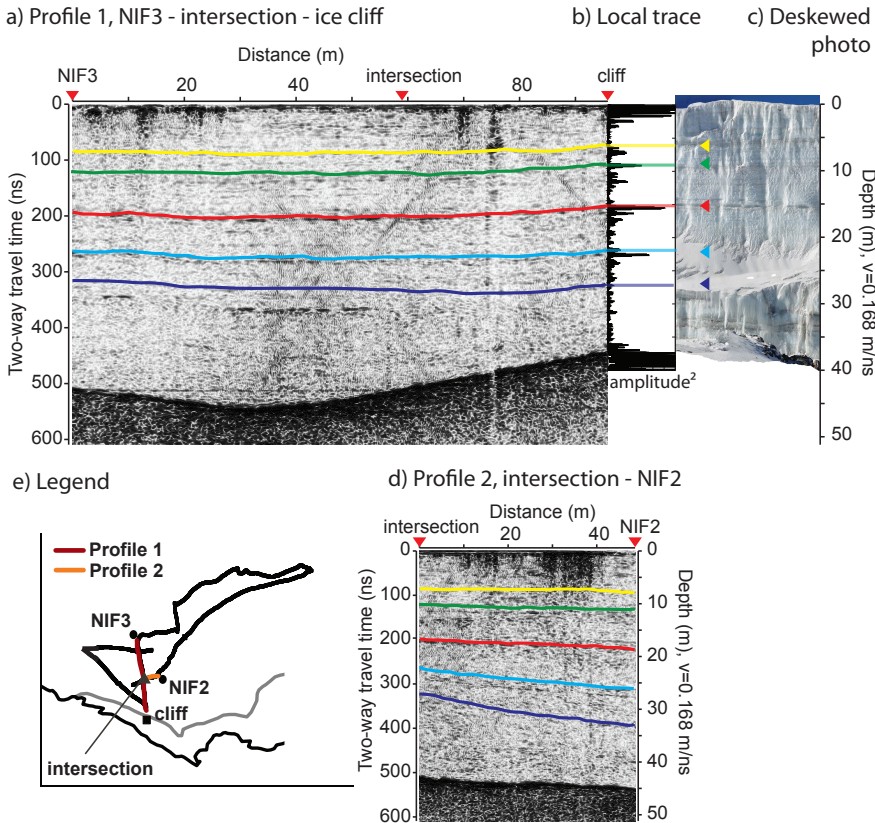

**Figure 4.** Direct comparison of processed GPR profiles with visible stratigraphy at the vertical wall. The location of the two GPR profiles is shown in e). Profile 1 extends to about 1 m before the ice cliff. Top row: Comparison with the full GPR profile (a). A local trace was extracted by averaging over 2 m at the end of profile 1 (b) and compared to a deskewed photo of the vertical wall (c). Note the reflection hyperbola of an open crevasse around 78 m in the profile. The colour-coded lines indicate manually picked internal reflectors, named IRH 1–5 and discussed in the text.

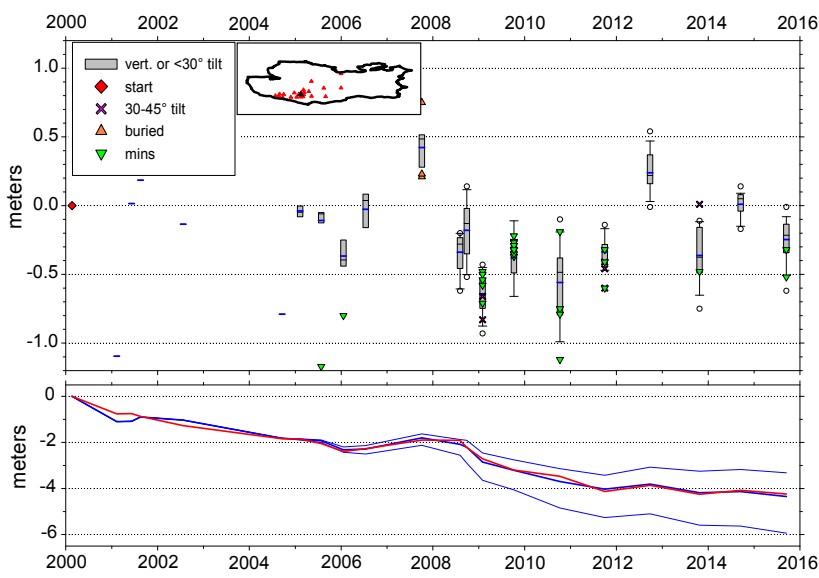

**Figure 5.** Ice surface elevation change at NIF derived from ablation stakes with at least two consecutive measurements (increasing from n=1 to n=19 stakes, in 2000 and 2015, respectively). The AWS and spatial coverage of stakes at NIF are shown next to the legend in the upper left (black and red triangles, respectively). In the top plot, grey box plots represent the distribution or change in ice height (median, quartiles) at vertical or near-vertical stakes ($< 30°$ tip; height measured along stake). Thick horizontal blue markers show the mean height change or height change values for only one measurement (i.e., 2001–2004). When the sample size is big enough, outliers are shown as black circles above/below the box plot whisker caps (90th and 10th percentiles). Stakes leaning $30 - 45°$, buried by accumulation, and those lying down due to ablation are shown as "x", orange and green inverted triangles, respectively; inverted triangles are thus minimum estimates of surface lowering. The lower plot shows cumulative ice height changes based on medians (thick blue line) and quartiles (thin blue) of the box plot dataset. Also shown (red line) is cumulative height change at the AWS; any snow overlying the ice is included in these heights (e.g., Feb. 2001), accounting for some of the apparent discrepancies.

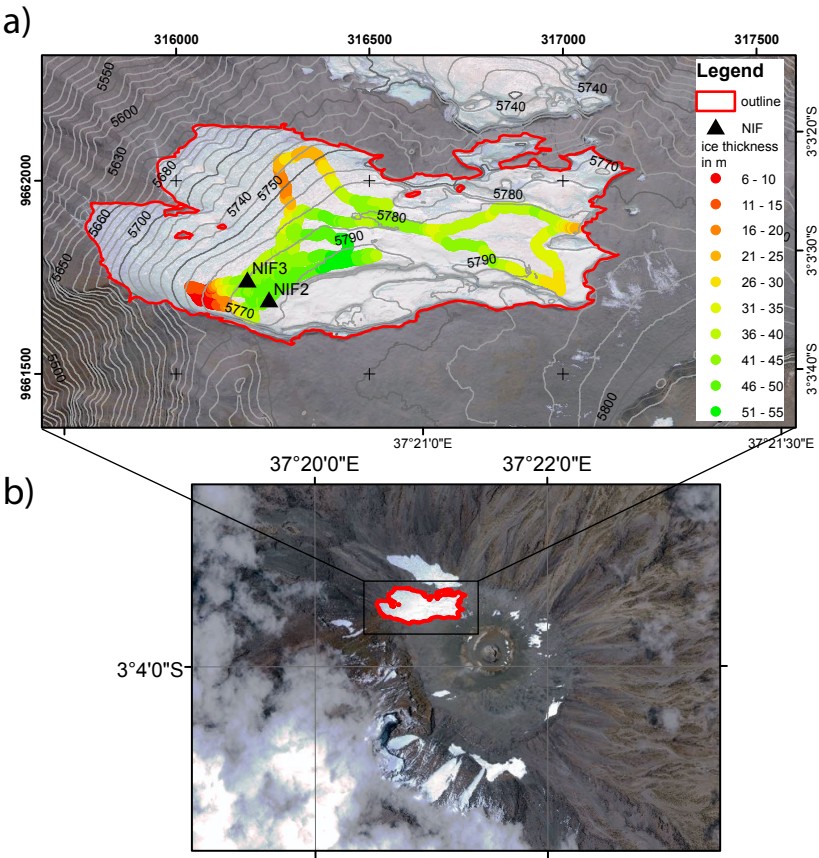

**Figure 6.** Ice thickness derived from the bed reflection in 100 and 200 MHz GPR profiles (a). The NIF outline is highlighted (red). Shown in b) is the NIF on an ortho-rectified GeoEye-1 satellite image acquired on 23 October 2012 (Sirguey and Cullen, 2014). Note the crater rim likely extending below the NIF. Coordinates are in UTM 37M.

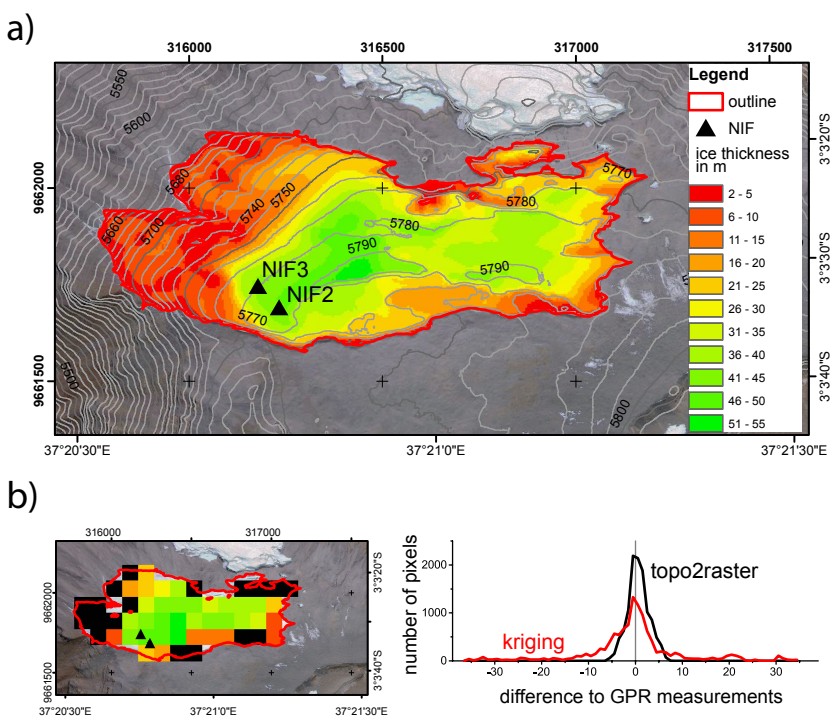

**Figure 7.** Interpolated ice thickness based on GPR profiles and the digital elevation model (a). The maximum ice thickness is around 53.5 m in approximately the center of the glacier, in association with the highest elevation. The bottom row b) shows the result of the alternative interpolation method by Kriging. Due to sparse coverage by GPR, kriging had to be applied at coarse resolution. Black squares indicate zero ice thickness. Shown on the right side is the value distribution of the difference calculated between GPR measured and interpolated ice thickness at GPR datapoints unused for interpolation (see text). Coordinates are in UTM 37M.

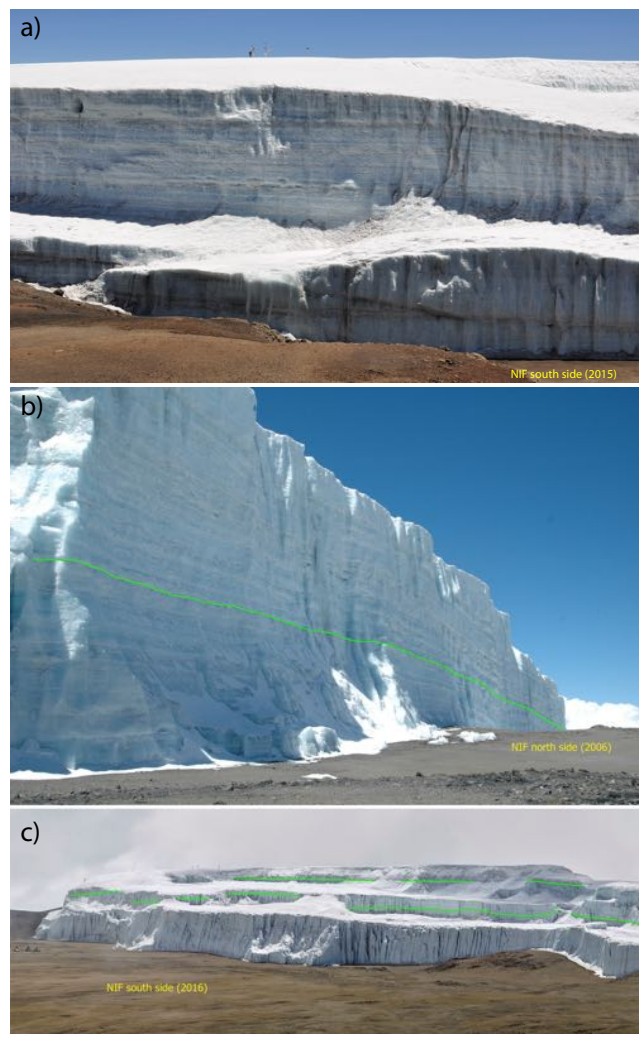

**Figure 8.** Multiple views of the stratigraphy at NIF's vertical walls. The top row a) shows the ice front near the vertical wall sampling site with a person standing next to the AWS. Note the exceptional inclined dark layer merging with another horizontal layer close to the crater surface. The ice front reveals distinct layers that are predominantly horizontal, seen both on the north and south side (b) and c), respectively). Examples of the distinct layers are highlighted as green lines.

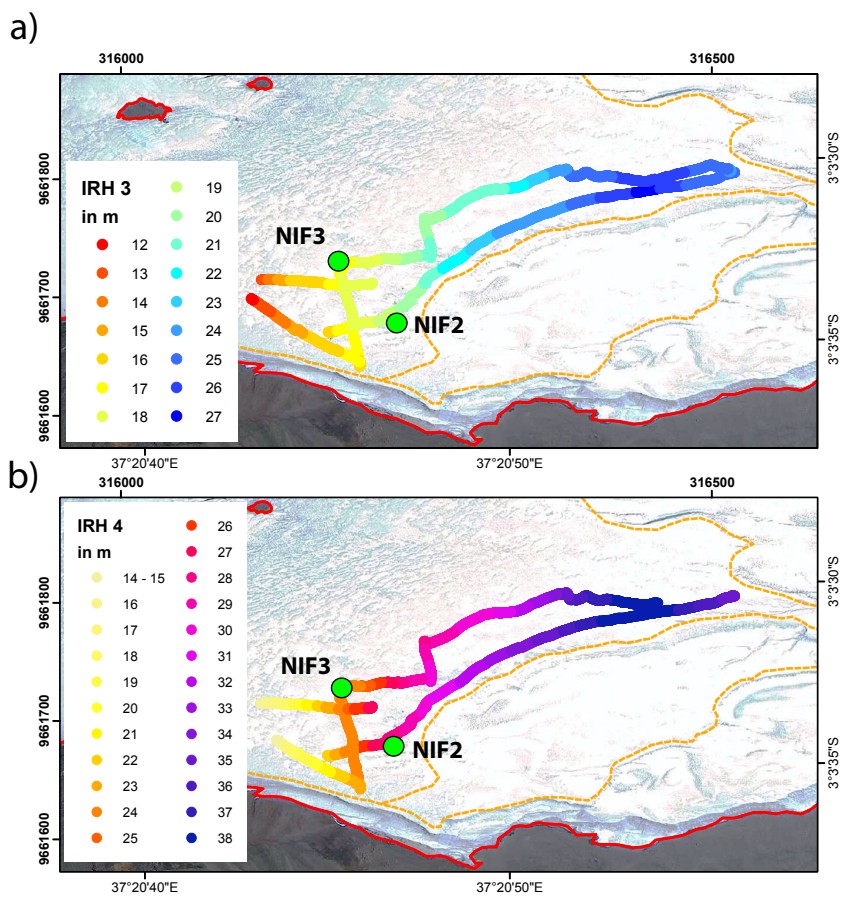

**Figure 9.** Tracing IRH in a closed course along all 200 MHz profile. Shown in the color-coded reflector depth of IRH 3 (a) and IRH 4 (b). Except for IRH 4 at the eastern end, reflectors can be connected at all intersections of two profiles. Coordinates are in UTM 37M.