# Peer review of "Ground-penetrating radar reveals ice thickness and undisturbed englacial layers at Kilimanjaro's Northern Ice Field"

_The Cryosphere, 2016_

## Referee Comment (RC1) · D. Samyn (Referee) · 12 Sep 2016

Bohleber et al. surveyed the Northern Ice Field of Kilimanjaro for reconstructing its bedrock topography, ice thickness and internal stratigraphy, using ground-penetrating radar (GPR) at various frequencies. Despite GPR being widely used in glaciology nowadays, this work is the first of its kind on Kilimanjaro, and therefore represents a novel approach in the exploration and investigation history of this mythical mountain. This study is well written, and I believe that the conclusions are scientifically sound and will contribute significantly to the future investigations of local, and other tropical, glacier recession dynamics.

As a general advice for improving this manuscript, I would suggest the authors to

strengthen their point where it is not stated carefully, or where the implications or interest for the scientific community are overlooked. These comments do not diminish the quality of this work though ; therefore I recommend publishing this paper with minor revisions as described below.

- Page 1, Line 7: "indicating an undisturbed internal stratigraphy within NIF's central flat area".

Whereas other statements of minor importance have been stressed more cautiously, I believe that this statement is too assertive and should be rephrased more carefully. Clearly some unknown uncertainty remains in this regard and, without drilling a new ice core between the former drilling sites and the edge ice cliff, without the result of the ice cliff dating work mentioned in the paper, and without carrying ice flow modelling investigations, no clear or solid information is available to certify that the internal stratigraphy is undisturbed. The influences on ice flow dynamics through time and space of, first, near-surface and internal meltwater and, second, fumaroles, still need to be better documented in order to fully appraise potential issues on the ice stratigraphical integrity. This comment also stands for the sentences on Page 9, Line 6 "We thus conclude that the internal stratigraphy within the NIF central flat area is generally undisturbed", and on Page 9, Line 32 "[. . .] revealed an undisturbed internal stratigraphy".

- Pages 4-5, "2.3 Uncertainty considerations" section

Here the vertical error in internal reflection horizons (IRH) tracking is discussed. How about the horizontal uncertainty related to the various GPR pulse triggering methods used (wheel, time, manual)? In other words, what is the horizontal extent of potential bedrock/stratigraphical discontinuities that the method used might omit while progressing on the glacier surface? This is of potential significance in regions of increased meltwater/fumarole activity, where electromagnetic coherency is more prone to disturbance.

- Page 5, Lines 12-14: "Assuming 0.3 m uncertainty in the length of the rope at 16 m

(mainly resulting from knots tied into the rope)".

From personal experience, the error stated seems rather low. In addition to the tied knots mentioned by the authors, the type of rope, its elasticity, and the mass of the dead weight at its end will certainly contribute. The uncertainty given here is therefore clearly a lower estimate.

- Page 7, Lines 21-22: "The low ice thickness is likely a result of the surface gradually sloping off towards the west outside the caldera. A distinct rise in the local GPR bedrock reflection appears where the location of the crater rim below the ice is suggested by satellite images (Figure 6, and small insert therein)".

The size of Fig. 6 inset is way too small to be able to observe this. This inset could certainly be resized to the dimensions of the main figure. In fact, it should, given the importance of the authors' point here.

- Page 7, Lines 23-24: "This finding implies that the local bedrock relief features may have affected past ice build up and decay through limiting exposure to solar radiation and wind".

I find this argument somewhat weak here – one would either need to check this limiting exposure effect with e.g. an insulation model, or provide more (visual?) details.

- Page 7, Lines 28-35: "Considering additionally the coarse resolution used in the kriging approach, we regard the values derived from this method with caution only. The estimates of total ice volume obtained from the Grid approach and DEM-only are $(12.0\pm0.3)$ and $(14.3\pm1.3)$ 106 m3, respectively. Evidently the main contribution to the difference in ice volume comes from different mean ice thickness values (using the 2012 surface area the mean ice thickness obtained from the Grid method gives a volume of $(12.3\pm0.3)$ 106 m3). The decrease in mean ice thickness suggested by the comparison of the two interpolation methods is not supported by surface height change measurements 2012–2015. Since both interpolation methods use the same

surface topography supplied by the DEM as input, the difference in mean ice thickness has to come from differences in determining subglacial bedrock. Consequently, the difference in ice volume estimates is not used to infer a rate of ice loss."

I wonder what is the added value of discussing the 'Kriging' method here, given its obvious flaws at such a low sampling resolution. There are various other interpolation techniques worth trying I think, that are not involving such a coarse resolution data grid.

- Page 7, Lines 31-33: "Evidently the main contribution to the difference in ice volume comes from different mean ice thickness values (using the 2012 surface area the mean ice thickness obtained from the Grid method gives a volume of (12.3±0.3) 106 m3)."

There should also be another source of error introduced in the volume calculations through the fact that ice cover area is simply multiplied by ice depth here, which is valid for a rectangular prism. The numbers given are thus upper estimates of the glacier volume.

- Page 8, Line 2: "we regard the ice volume estimate of the Grid method as most accurate".

As mentioned for Page 7, Lines 28-35, this statement is somewhat trivial here.

- Page 8, Lines 12-13: "It is worth noting that the vertical cliffs show instances of tilted and converging layers in close proximity to bedrock".

Instead of 'converging' layers, the pattern in question rather looks in my opinion, from visual inspection of Fig. 8, like a layer from which another layer is swelling as a result of a rheological discontinuity (e.g. localized shearing), as often occurs at the margin of glaciers. This has potential implications not only for the detection of deep reflectors as stated by the authors, but also for the integrity of the ice layering. This comment, which I believe needs to be discussed in the manuscript, highlights my former comment on Page 1, Line 7 regarding the authors' rationale and uncertainty analysis on the argued 'undisturbed internal stratigraphy'.

- Page 8, Lines 14-15: "[. . .] where ice thickness decreases rapidly due to the crater rim".

I do not think that the presence of the crater rim is the only reason for this 'ice thickness decrease'. In the case where, say after a period of increased accumulation rate, more ice would flow towards the ice rim, ice thickness could in fact increase as a result of the blocking effect by the rim. In the case discussed by the authors, it is probably the conjunction of the rim vicinity and stagnant flow that causes the ice to reduce locally in thickness.

- Page 8, Lines 20-23: "It is plausible that the according change in the electrical conductivity of the ice layer produces a strong reflector seen in the GPR data (Sold et al., 2015). Accordingly, this strongly suggests dust layers being a main physical cause of IRH at NIF. Thompson et al. (2002) and Gabrielli et al. (2014) report visible dust layers in the NIF2 and NIF3 ice cores".

If the change in electrical conductivity expected from the ammonium and chloride documented by Thompson et al. (2002) results indeed from dust layers, a consequent change in ice crystal texture should also be expected, given the retardation effects of micro-particles on grain boundary migration and recrystallization. IRH might thus represent "iso-chemical" AND "iso-crystalline" reflectors.

- Page 8, Line 33-Page 9, Line 8: discussion on IRH 1-5 tracking.

This discussion could be somewhat improved and made much clearer with the use, for instance, of a table giving (1) the expected depth of these horizons from previous ice cores, and (2) their depth detected by GPR. The total lengths between the drilling sites, the ice cliff, and the locations where the IRH tracks are lost would also be helpful in order to appraise the layer continuity/extension.

The ratio of vertical distances separating the IRH discussed at various locations would also help evaluating the vertical stratigraphical dilatation/shrinking along the studied

profiles.

- Page 9: Lines 9-19: discussion on continuous layering.

It is not clear, from this paragraph, where the authors want to lead the reader. It is only after reading the Conclusion section that one is able to get the authors' point regarding the importance of stratigraphical continuity between the former drill sites and the ice cliff: they are concerned about the possibility to efficiently and confidently relate the results from former ice cores to the results of the ice dating work along the ice cliff. This concern is totally justified here, and should be wrapped up more tightly in this section.

- Page 9, Lines 15-19: "Although qualitatively going in the same direction as the adjustment of the NIF2 and NIF3 stable isotope records (i.e. in comparison with Figure 2 in Thompson et al. (2002)), tracing IRH between NIF2 and NIF3 suggests tie points that are systematically at greater depth in NIF3 as compared to the ice core stable isotope matching."

Do the authors have an idea about why the ice stratigraphy is stretched at NIF3? Differences in accumulation cannot really be invoked here given the small distance between both NIF2 and NIF3 sites. Ice flow would probably play a role, which is difficult to determine without ice flow modelling though.

- Page 9, Lines 26-29: "Hence our GPR profiles demonstrate a highly heterogeneous presence of meltwater near the surface, apparently a wide-spread feature at NIF related to spatial and temporal variability in surface characteristics and processes (Hardy, 2011). This finding is of relevance for any new ice core drilling efforts at NIF in the future, and an important consideration for energy and mass balance modelling efforts."

Although this section is called "Effects of near-surface meltwater", these effects are not really discussed. The authors are only referring to this issue as "of relevance for". I suggest that they either discuss this important issue more thoroughly, or suppress this

section. This comment also applies to Lines 11-12 in the Conclusion section.

---

## Referee Comment (RC2) · Anonymous Referee #2 · 19 Sep 2016

This manuscript presents the GPR data collected on Kilimanjaro's Northern Ice Field for the first time and estimate the total ice volume as of September 2015. Also, the integrity of internal reflecting horizons for the majority of the NIF is clearly established here, opening possibilities for future studies such as extending the depth-age relationship obtained from ice cores to reconstruct the historical change of the NIF. The manuscript is well structured and concise. I have only a few minor comments on uncertainty analysis, discussion of results in light of previous studies, editorial comments to clarify the writing, and the size of figures and some text embedded in them. I recommend this manuscript for publication in The Cryosphere after a minor revision.

Specific comments

[Figure]

Section 2.3: There is no discussion about the horizontal uncertainty that could arise from the determination of from where the pulse is returned, for example. Please add some discussion of the horizontal uncertainty.

P4, L27-28: I'm not totally clear on how you calculated the combined uncertainties here. These uncertainty components are independent of each other so I think the proper way to combine the uncertainties in this case is by the root sum of squares. So for the IRH and the bedrock reflection at 200 MHz, they would be sqrt(2.5^2+4^2)=4.7 ns and sqrt(2.5^2+8^2)=8.4 ns, respectively.

P5, L4-5: The total uncertainties for the IRH and bedrock depths would change depending on how you combine different uncertainty components as per the comment above. Please check the final number and change as needed.

P5, L12-13: It is difficult to assess if 0.3 m is appropriate for the uncertainty of the rope length because there is no explanation as to how knots would lead to this number. In addition, I would expect some stretching of the rope unless you specifically chose a static rope with minimal stretching.

P5, L13-14: Why could you neglect potential effects from the image stitching and deskewing routines? Are there any references to justify this?

P7, L1: What is the significance of the "large bedrock inclination"? Is this related to one of the components of the uncertainty, namely losing track of coherent phase? Otherwise, this whole sentence seems to imply that there was in fact a component of uncertainty other than the two you discussed in section 2.3 but you got away with considering only the two by chance. Please clarify.

P7, L14-16: I don't agree that the observed mismatch could be attributed to the combined uncertainty. My interpretation of this statement is that your analysis of the combined uncertainty is wrong, which would require you to revise section 2.3. I don't think that is the case. It seems as though the mismatch could be largely due to the spatial

and possibly the temporal variability (?) of the bottom melting caused by fumarole ac-
tivities, which are not well documented so you are not able to quantify it, and a potential
uncertainty in the core length.

P8, L29-30: The discrepancy between your finding and the interpretation of Thompson
et al. is significant. This warrants further discussions, at least further explain what
Thompson et al.'s interpretation is and more details on how your result questions their
interpretation.

Technical corrections

P2, L28: The use of the word "employed" is awkward. Change to "GPR has also been
used..."

P2, L32: Add "e.g.," to the references because these might not be the only studies that
used GPR on tropical glaciers.

P2, L32-33: "to our knowledge the study presented here..." should be '"to our knowl-
edge this is the first time a GPR was used at Kilimanjaro's NIF."

P3, L3-5: The sentence "Although not further discussed..." seems unnecessary if not
discussed at all in this manuscript.

P3, L5-6: The sentence should be changed to "We estimate the total ice volume
presently remaining at NIF by spatially extrapolating the GPR-derived ice thickness."

P3, L8: Change "while" to "and".

P3, L9-10: You've defined the abbreviation already so use "IRH".

P3, L14: Change "as well as" to "and".

P3, L18: Change "employed" to "used".

P3, L18: Change "Technical settings of the setups" to "Details of the technical settings".

P3, L23: Change "The spatial coverage that could be achieved was constrained by" to

"The spatial extent of the GPR survey was constrained by ".

P3, L24: Change "employ" to "use".

P3, L27: Change "800 MHz profiles were not found to provide" to "800 MHz profiles did not provide".

P4, L5: I think "Post-processing of GPR data" reads better as a subsection heading.

P4, L6: "We used the standard routines to process the GPR data including ..."

P4, L9-11: The use of "while" in the sentence "We employed ..." is not appropriate so the sentence should be divided, with the first sentence ending after "5 traces" and the second sentence starting with "For the electromagnetic ...".

P4, L20: "Major contributions to the uncertainty in depth..."

P4, L21: Change "connected to" to "related to".

P4, L25: Change "loosing" to "losing".

P4, L26-27: You don't need the parenthesis.

P4, L29: Delete "relative difference".

P5, L8-9: Change "A 200 MHz CO-profile running towards the vertical wall extends to about one meter distance from the cliff" to "The 200 MHz CO-profile running towards the ice cliff ends within one meter from the cliff".

P5, L9: Change "The cliff height of the wall" to "The height of the ice cliff".

P5, L16: "In order to derive distributed ice thickness" to "To derive the ice-thickness distribution over the NIF", and remove the later "over the NIF".

P5, L16-17: Change "the previously developed approach by Fischer (2009), in interpolating" to "the approached previously developed by Fischer (2009), first interpolating".

P5, L21: "very high resolution" is subjective so remove "very".
P5, L22: No hyphen is needed for surface altitude.

P5, L33: Change "We derived an estimate" to "We estimated".

P6, L3: Change "In order to estimate the expected loss on surface area" to "To estimate the surface area lost".

P6, L14: Change "comprises" to "includes".

P6, L18: Change "reflectors from internal layers" to "internal reflectors".

P6, L19: Remove "very".

P6, L28: You don't need parentheses around the description of locations.

P6, L30: Delete ", however".

P7, L4: Remove "value".

P7, L13: "more or less" is ambiguous so remove.

P7, L17: Change "The interpolation of ice thickness" to "The interpolated ice thickness distribution".

P7, L28: Change "Considering additionally" to "In addition, considering".

P7, L28-29: Change "regard the values derived from this method with caution only" to "interpret the ice thickness derived from this method with caution."

P8, L27: Change "large layer" to "thick layer".

P8, L29: Change "interpret" to "interpreted".

P8, L29: Remove "in depth".

P8, L30-32: It isn't totally clear whether "these findings" refer to your findings or those of Thompson et al. (I assume the former). Rewrite to clarify this.

P8, L30: Change "it seems worth" to "it is".

P9, L7: Change "near-bedrock ice parts" to "ice just above the bedrock".

P9, L28-29: Briefly explain why this finding is relevant for new ice core drilling and energy and mass balance modeling.

P9, L31: Change "estimation" to "estimate".

P10, L2: Change "can be" to "were".

This is something you could sort out with TC's but I think figures are a little too small in general. Please pay particular attention to the size of texts embedded in each figures and make sure they are legible without blowing up on a computer screen. Labels of site and profile names in Figure 1, and legends in Figures 5 and 7 are particularly difficult to read.

Figures 1, 2 and 9: Label the top and bottom rows as (a) and (b), respectively, and refer to them accordingly in captions.

---

## Author Comment (AC1) · 11 Nov 2016

**"Ground-penetrating radar reveals ice thickness and undisturbed englacial**

**layers at Kilimanjaro's Northern Ice Field" by Pascal Bohleber et al.**

- Response to reviews and revised manuscript -

***General Remarks:*** *All line numbers in "Changes to manuscript" refer to the revised*

*version. Changes in the corresponding pdf of the revised manuscript are highlighted in*

*red.*

*Author's responses to the referee's comments are in blue.*

*All new references used in this text here can be found in the revised manuscript.*

**Response to referee #1 (Denis Samyn) posted on Sept. 12th 2016**

Bohleber et al. surveyed the Northern Ice Field of Kilimanjaro for reconstructing its bedrock topography, ice thickness and internal stratigraphy, using ground- penetrating radar (GPR) at various frequencies. Despite GPR being widely used in glaciology nowadays, this work is the first of its kind on Kilimanjaro, and therefore represents a novel approach in the exploration and investigation history of this mythical mountain. This study is well written, and I believe that the conclusions are scientifically sound and will contribute significantly to the future investigations of local, and other tropical, glacier recession dynamics.

As a general advice for improving this manuscript, I would suggest the authors to strengthen their point where it is not stated carefully, or where the implications or interest for the scientific community are overlooked. These comments do not diminish the quality of this work though; therefore I recommend publishing this paper with minor revisions as described below.

We thank the referee for a very thorough review, we appreciate the helpful suggestions and comments.

**Referee comment**

- Page 1, Line 7: "indicating an undisturbed internal stratigraphy within NIF's central flat area".

Whereas other statements of minor importance have been stressed more cautiously, I believe that this statement is too assertive and should be rephrased more carefully. Clearly some unknown uncertainty remains in this regard and, without drilling a new ice core between the former drilling sites and the edge ice cliff, without the result of the ice cliff dating work mentioned in the paper, and without carrying ice flow modelling investigations, no clear or solid information is available to certify that the internal stratigraphy is undisturbed. The influences on ice flow dynamics through time and space of, first, near-surface and internal meltwater and, second, fumaroles, still need to be better documented in order to fully appraise potential issues on the ice stratigraphical integrity. This comment also stands for the sentences on Page 9, Line 6 "We thus conclude that the internal stratigraphy within the NIF central flat area is generally undisturbed", and on Page 9, Line 32 "[...] revealed an undisturbed internal stratigraphy".

We believe the presence of spatially continuous internal reflection horizons in the GPR profiles stem from an uninterrupted, spatially coherent layering within the NIF plateau area, which is one of the central findings of our study. Limitations to this finding apply to the near-surface sections where noise associated with meltwater hampers tracing reflections, as well as to the near-basal sections where strong continuous reflections are not detected. Our main point is that the coherent stratigraphy in the 200 MHz profiles does not provide any evidence for deformed (overturned, interrupted) layers. Based on the referee's comment we understand that the general use of the term "undisturbed stratigraphy" can be misinterpreted. Hence we decided to replace the term "undisturbed stratigraphy" with "uninterrupted, spatially coherent internal layering ". We also clarified on the depth restriction of the tracing of IRH in the abstract.

We agree with the referee that additional information regarding the influence of
meltwater percolation (especially on the cm-scale chemical stratigraphy in ice
cores), as well as investigating basal fumarole activity would be helpful for an even
more refined assessment of the stratigraphy at NIF and regard this a helpful
suggestion for future research.
**Changes to manuscript:**
• Page 1, Line 7: "indicating an uninterrupted, spatially coherent internal
layering "
• Page 1, Line 8: "We show that, at least for the upper 30 m, it is possible to
follow isochrone layers between two former NIF ice core drilling sites and a
sampling site on NIF's vertical wall."
• Page 9, Line 16-17: "generally composed of uninterrupted, spatially coherent
layers"
• Page 10, Line 19-20: "an internal stratigraphy made up of an uninterrupted,
spatially coherent layering.
**Referee comment**
- Pages 4-5, "2.3 Uncertainty considerations" section
Here the vertical error in internal reflection horizons (IRH) tracking is discussed.
How about the horizontal uncertainty related to the various GPR pulse triggering
methods used (wheel, time, manual)?  In other words, what is the horizontal extent
of potential bedrock/stratigraphical discontinuities that the method used might
omit while progressing on the glacier surface?   This is of potential significance in
regions of increased meltwater/fumarole activity, where electromagnetic coherency
is more prone to disturbance.
We thank the referee for this suggestion and have now added a short discussion of
the horizontal resolution in section 2.3 "uncertainty considerations". In essence we are following earlier studies by Welch et al. (1998) and Yilmaz (1987), who showed that for properly migrated radargrams the horizontal resolution becomes lambda/2, independent of reflector depth. In data acquisition we took care to avoid spatial aliasing by collecting traces less than one quarter wavelength apart.

**Changes to manuscript:**

• Page 5, Line 6 ff.: " Shot distances in data acquisition... "

**Referee comment**

- Page 5, Lines 12-14: "Assuming 0.3 m uncertainty in the length of the rope at 16 m (mainly resulting from knots tied into the rope)".

From personal experience, the error stated seems rather low.  In addition to the tied knots mentioned by the authors, the type of rope, its elasticity, and the mass of the dead weight at its end will certainly contribute. The uncertainty given here is therefore clearly a lower estimate.

We agree with the referee and have added text to clarify that we are regarding this uncertainty as merely a lower estimate.

**Changes to manuscript:**

• Page 5, Lines 17: "To derive a lower estimate of uncertainty..."

**Referee comment**

- Page 7, Lines 21-22: "The low ice thickness is likely a result of the surface gradually sloping off towards the west outside the caldera.  A distinct rise in the local GPR bedrock reflection appears where the location of the crater rim below the ice is suggested by satellite images (Figure 6, and small insert therein)".

The size of Fig. 6 inset is way too small to be able to observe this. This inset could certainly be resized to the dimensions of the main figure. In fact, it should, given the importance of the authors' point here.

We took care to resize the insert in order to aid better visual recognition of the satellite image. As a general remark, we have also tried to improve the readability of all of the figures by increasing font size etc.

**Changes to manuscript:**

• Figure 6: Resized insert to full size

**Referee comment**

- Page 7, Lines 23-24: "This finding implies that the local bedrock relief features may have affected past ice build up and decay through limiting exposure to solar radiation and wind".

I find this argument somewhat weak here – one would either need to check this limiting exposure effect with e.g. an insulation model, or provide more (visual?)

details.

We did not intend to make this argument based on our findings alone. Instead, we wanted to point out the detection of the subglacial crater rim in context of the previous study of Kaser et al. (2010) who suggested that local bedrock relief features may have affected past ice build up and decay through limiting exposure to solar radiation and wind. We have changed the sentence to clarify accordingly.

**Changes to manuscript:**

• Page 7, Lines 34 ff.: "This finding supports the idea that local bedrock relief features may have affected past ice build up and decay through limiting exposure to solar radiation and wind (Kaser et al., 2010)."

**Referee comment**

- Page 7, Lines 28-35: "Considering additionally the coarse resolution used in the kriging approach, we regard the values derived from this method with caution only.

The estimates of total ice volume obtained from the Grid approach and DEM-only are (12.0±0.3) and (14.3±1.3) $10^6$ m³, respectively. Evidently the main contribution to the difference in ice volume comes from different mean ice thickness values (using the 2012 surface area the mean ice thickness obtained from the Grid method gives a volume of (12.3 ± 0.3) $10^6$ m³). The decrease in mean ice thickness suggested by the comparison of the two interpolation methods is not supported by surface height change measurements 2012–2015. Since both interpolation methods use the same surface topography supplied by the DEM as input, the difference in mean ice thickness has to come from differences in determining subglacial bedrock.

Consequently, the difference in ice volume estimates is not used to infer a rate of ice loss."

I wonder what is the added value of discussing the 'Kriging' method here, given its obvious flaws at such a low sampling resolution. There are various other interpolation techniques worth trying I think, that are not involving such a coarse resolution data grid.

Our intention was to include the 'Kriging' method as an alternative spatial interpolation routine that uses the GPR based derived ice thickness profiles only.

The coarse spatial resolution is an immediate consequence of the sparse spatial coverage of the GPR profiles over the NIF. In this respect, a finer mesh-type array of profiles would have been desirable but was not feasible due to time and issues related to surface roughness. We agree that the results of the 'Kriging' routine provide less detail in comparison with the DEM-based and 'Grid' interpolation scheme. We are already stating in the manuscript that the 'Kriging' results are regarded with caution only. In the end we decided to leave the 'Kriging' results in
the text in order to illustrate to the reader the benefit of the GPR-DEM combined
interpolation approach. We have changed the text to make this intention more clear.
While a detailed analysis of the result of various interpolation models and
techniques is far beyond the scope of this paper, the IACS working group on ice
thickness has just submitted a paper on this topic with a large sample of glaciers of
various types ("ITMIX experiment"). This promises much greater insight as
compared to investigating one glacier only. As the data of our study will be
submitted to GlaThiDA 3.0, the data will also be available for validation of a
potential second ITMIX experiment.
**Changes to manuscript:**
• Page 6, Lines 19-21: "Although clearly suffering from these restrictions..."
**Referee comment**
- Page 7, Lines 31-33: "Evidently the main contribution to the difference in ice
volume comes from different mean ice thickness values (using the 2012 surface area
the mean ice thickness obtained from the Grid method gives a volume of (12.3
± 0.3) $10^6$ m$^3$)."
There should also be another source of error introduced in the volume calculations
through the fact that ice cover area is simply multiplied by ice depth here, which is
valid for a rectangular prism.  The numbers given are thus upper estimates of the
glacier volume.
We agree that using the mean ice thickness multiplied by the total surface area can
only give an estimate. Calculating the volume by multiplying area by height luckily
works for every prism (and not just rectangular ones). Using the areal mean height
(including its uncertainty) should avoid a systematic overestimation. What we
intend to point out in the above mentioned is the fact that the dominant cause for the difference in ice volume estimates between the Grid and DEM-only approach is
due to different ice thickness values, as opposed to the additional contribution of
different surface area. We have changed the sentence to clarify.
**Changes to manuscript:**
• Page 8, Line 7-8: "The main contribution to the difference in ice volume
comes from different mean ice thickness values as opposed to surface area"
**Referee comment**
- Page 8, Line 2: "we regard the ice volume estimate of the Grid method as
most accurate".
As mentioned for Page 7, Lines 28-35, this statement is somewhat trivial here.
In this instance, we are not referring anymore to a comparison with the coarse
interpolation based on 'Kriging', but compare the DEM-based and the DEM+GPR-
combined approach. The fact that GPR introduces additional constraints may indeed
sound trivial to the reader. However, we felt it was necessary to be clear about
which ice volume estimate is regarded as the final and most reliable estimate. We
have slightly modified our wording in this regard.
**Changes to manuscript:**
• Page 8, Lines 13-14: "Integrating both the DEM and GPR as constraints, the
Grid method provides the most reliable ice volume estimate"
**Referee comment**
- Page 8, Lines 12-13: "It is worth noting that the vertical cliffs show instances of
tilted and converging layers in close proximity to bedrock".

Instead of 'converging' layers, the pattern in question rather looks in my opinion, from visual inspection of Fig. 8, like a layer from which another layer is swelling as a result of a rheological discontinuity (e.g. localized shearing), as often occurs at the margin of glaciers. This has potential implications not only for the detection of deep reflectors as stated by the authors, but also for the integrity of the ice layering. This comment, which I believe needs to be discussed in the manuscript, highlights my former comment on Page 1, Line 7 regarding the authors' rationale and uncertainty analysis on the argued 'undisturbed internal stratigraphy'.

We thank the referee for pointing out this additional hypothesis and we have integrated this point into our discussion. However, we believe that this stratigraphic convergence is an ablation feature rather than due rheology, as localized shearing appears evident only near the snout of the steepest slope glaciers, and features such as that shown in Fig. 8 occur elsewhere on Kilimanjaro glaciers, particularly those on the south side.

**Changes to manuscript:**

•   Page 8, Lines 25-28:  "We believe that this stratigraphic convergence is an ablation feature rather than due rheology (e.g. localized shearing at the glacier margin), as localized shearing appears evident only near the snout of the steepest slope glaciers, and features such as that shown in Figure 8 occur elsewhere on Kilimanjaro glaciers, particularly on the south side."

**Referee comment**

- Page 8, Lines 14-15:  "[...] where ice thickness decreases rapidly due to the crater rim".

I do not think that the presence of the crater rim is the only reason for this 'ice thickness decrease'. In the case where, say after a period of increased accumulation rate, more ice would flow towards the ice rim, ice thickness could in fact increase as a result of the blocking effect by the rim.  In the case discussed by the authors, it is probably the conjunction of the rim vicinity and stagnant flow that causes the ice to reduce locally in thickness.

We appreciate this input by the referee. We were not trying to say the crater rim is the original cause of the decrease in ice thickness, but were simply referring to the situation as of today mapped by our GPR profiles. We have modified the wording to clarify. That said we are not aware of any direct evidence nor published accounts of ice flow at NIF.

**Changes to manuscript:**

• Page 8, Lines 29-30: "... in the part of the profiles showing decreasing ice thickness and gradual slope in the bedrock, likely the crater rim."

**Referee comment**

- Page 8, Lines 20-23: "It is plausible that the according change in the electrical con- ductivity of the ice layer produces a strong reflector seen in the GPR data (Sold et al.,

2015).  Accordingly, this strongly suggests dust layers being a main physical cause of IRH at NIF. Thompson et al. (2002) and Gabrielli et al. (2014) report visible dust layers in the NIF2 and NIF3 ice cores".

If the change in electrical conductivity expected from the ammonium and chloride documented by Thompson et al.  (2002) results indeed from dust layers,  a consequent change in ice crystal texture should also be expected,  given the retardation effects of micro-particles on grain boundary migration and recrystallization.  IRH might thus represent "iso-chemical" AND "iso-crystalline"

reflectors.

This is an interesting suggestion and we agree that the known interaction between impurities and ice texture evolution can be expected also at NIF. IRH caused by ice texture are linked to the anisotropic dielectric properties of ice. Hence, a change in ice texture (i.e. grain size) is not sufficient for an IRH to occur, but would also need to go along with a systematic local anisotropy in crystal orientation. In turn, this would also imply a dependency on the electric polarisation of the GPR pulse. We have not observed a change in reflectors at points were we have almost perpendicular intersections of GPR profiles (e.g. point "intersection" in Fig. 4).

Although we cannot entirely rule out the possibility for a contribution of crystal orientation to individual IRH, we feel that the change in ice chemistry at the large dust bands is certainly strong enough to explain all major IRHs discussed here.

**Changes to manuscript:**

No change necessary.

**Referee comment**

- Page 8, Line 33-Page 9, Line 8: discussion on IRH 1-5 tracking.

This discussion could be somewhat improved and made much clearer with the use, for instance, of a table giving (1) the expected depth of these horizons from previous ice cores, and (2) their depth detected by GPR. The total lengths between the drilling sites, the ice cliff, and the locations where the IRH tracks are lost would also be helpful in order to appraise the layer continuity/extension.

The ratio of vertical distances separating the IRH discussed at various locations would also help evaluating the vertical stratigraphical dilatation/shrinking along the studied profiles.

Except for IRH 5, which appears to clearly correspond to the exceptionally large dust layer found in the NIF3 ice core, the derivation of expected IRH depths based on the impurity profiles of the ice cores remains ambiguous (except of the expected depth of the known dust horizons which we have already included in the text).

However, we have followed the referee's suggestion and added to Table 3 a column for horizontal distances (in correspondence to Figure 4). We also now include the relative depth for each IRH in Table 3 to aid evaluating the vertical stratigraphical dilatation/shrinking.

**Changes to manuscript:**

•  Modified Table 3 to include horizontal distances and relative depths of IRH.

**Referee comment**

- Page 9: Lines 9-19: discussion on continuous layering.

It is not clear, from this paragraph, where the authors want to lead the reader. It is only after reading the Conclusion section that one is able to get the authors' point regarding the importance of stratigraphical continuity between the former drill sites and the ice cliff:  they are concerned about the possibility to efficiently and confidently relate the results from former ice cores to the results of the ice dating work along the ice cliff. This concern is totally justified here, and should be wrapped up more tightly in this section.

We thank the reviewer for pointing this out and have added text to reiterate here in modified form what is said in the Conclusions.

**Changes to manuscript:**

•  rewrote paragraph on Page 9, starting Line 19.

**Referee comment**

- Page 9, Lines 15-19: "Although qualitatively going in the same direction as the adjustment of the NIF2 and NIF3 stable isotope records (i.e. in comparison with

Figure 2 in Thompson et al. (2002)), tracing IRH between NIF2 and NIF3 suggests tie points that are systematically at greater depth in NIF3 as compared to the ice core stable isotope matching."

Do the authors have an idea about why the ice stratigraphy is stretched at NIF3?

Differences in accumulation cannot really be invoked here given the small distance between both NIF2  and  NIF3  sites.  Ice flow would probably play a role, which is difficult to determine without ice flow modelling though.

We do not have a conclusive explanation for this situation, and at this time can only note that the difference in relative depths seems to be predominant at lower depths (which becomes more evident by the revised version of Table 3 now). It also seems worth noting in this context that, as a general case at NIF, the visible dust bands on the vertical walls appear to vary in their relative depth. We agree with the referee that systematic differences in accumulation appear unlikely and, as stated previously, question whether ice flow could be involved in altering the stratigraphy of this thin, nearly-horizontal section of the glacier.

**Changes to manuscript:**

• Changes in Table 3.

• Additional clarification in paragraph on page 9, starting line 26.

**Referee comment**

- Page 9, Lines 26-29: "Hence our GPR profiles demonstrate a highly heterogeneous presence of meltwater near the surface, apparently a wide-spread feature at NIF re- lated to spatial and temporal variability in surface characteristics and processes (Hardy,2011). This finding is of relevance for any new ice core drilling efforts at NIF

in the future, and an important consideration for energy and mass balance modelling efforts."

Although this section is called "Effects of near-surface meltwater", these effects are not really discussed.  The authors are only referring to this issue as "of relevance for".  I suggest that they either discuss this important issue more thoroughly, or suppress this section. This comment also applies to Lines 11-12 in the Conclusion section.

We agree that this is an important finding, although not in the original focus of our work. Hence we followed the referee's suggestion and have elaborated more on the relevance to future ice core drillings as well as modelling efforts.

**Changes to manuscript:**

• Page 10, Lines 13-16: "...suggesting that chemical and isotopic records of the upper 10~m or more could be potentially corrupted by meltwater. The wide- spread presence of near-surface meltwater also needs to be considered in future energy and mass balance modelling efforts. Further quantifying the generation and evolution of the near-surface meltwater distribution points to important future research questions at NIF.

---

## Author Comment (AC2) · 11 Nov 2016

**"Ground-penetrating radar reveals ice thickness and undisturbed englacial**

**layers at Kilimanjaro's Northern Ice Field" by Pascal Bohleber et al.**

- Response to reviews and revised manuscript -

***General Remarks:*** *All line numbers in "Changes to manuscript" refer to the revised*

*version. Changes in the corresponding pdf of the revised manuscript are highlighted in*

*red.*

*Author's responses to the referee's comments are in blue.*

*All new references used in this text here can be found in the revised manuscript.*

**Response to anonymous referee #2 posted on Sept. 19th 2016**

This manuscript presents the GPR data collected on Kilimanjaro's Northern Ice Field for the first time and estimate the total ice volume as of September 2015. Also, the integrity of internal reflecting horizons for the majority of the NIF is clearly established here, opening possibilities for future studies such as extending the depth-age relationship obtained from ice cores to reconstruct the historical change of the NIF. The manuscript is well structured and concise. I have only a few minor comments on uncertainty analysis, discussion of results in light of previous studies, editorial comments to clarify the writing, and the size of figures and some text embedded in them. I recommend this manuscript for publication in The Cryosphere after a minor revision.

Thank you very much for your review and helpful suggestions!

**Specific comments**

**Referee comment**

Section 2.3: There is no discussion about the horizontal uncertainty that could arise from the determination of from where the pulse is returned, for example. Please add some discussion of the horizontal uncertainty.

This point was noted by both referees and we took care to add information regarding the horizontal resolution in section 2.3 "uncertainty considerations".

**Changes to manuscript:**

- Page 5, Line 6 ff.: " Shot distances in data acquistion... "

**Referee comment**

P4, L27-28: I'm not totally clear on how you calculated the combined uncertainties here. These uncertainty components are independent of each other so I think the proper way to combine the uncertainties in this case is by the root sum of squares. So for the IRH and the bedrock reflection at 200 MHz, they would be sqrt(2.5^2+4^2)=4.7ns and sqrt(2.5^2+8^2)=8.4 ns, respectively.

Thank you for pointing this out. The values of 6 and 9 ns were erroneously reported for 200 MHz but belong to 100 MHz. We have corrected the text accordingly and changed the values where needed (we rounded to full ns and m, respectively).

**Changes to manuscript:**

- Page 4, Lines 25-26: Changed values and explicitly noted that the root sum of
  squares was used.

**Referee comment**

P5, L4-5: The total uncertainties for the IRH and bedrock depths would change depending on how you combine different uncertainty components as per the comment above. Please check the final number and change as needed.

Thank you, we have corrected the values, see comment above.

**Changes to manuscript:**

• Page 5, Lines 2-3: Changed values accordingly.

**Referee comment**

P5, L12-13: It is difficult to assess if 0.3 m is appropriate for the uncertainty of the rope length because there is no explanation as to how knots would lead to this number. In addition, I would expect some stretching of the rope unless you specifically chose a static rope with minimal stretching.

We made an effort to estimate at first order how much the length of the rope changes based on the knots. We agree that some rope streching can be expected and have now clarified that we regard our estimate as a lower limit of uncertainty only.

**Changes to manuscript:**

• Page 5, Lines 17 ff.: "To derive a lower estimate of uncertainty..."

**Referee comment**

P5, L13-14: Why could you neglect potential effects from the image stitching and deskewing routines? Are there any references to justify this?

We thank the referee for pointing this out and have now included discussing the uncertainty of image stitching and deskewing routines. Although we are unable to come up with a quantified estimate we believe this contribution is negligible and have add references to justify this.

**Changes to manuscript:**

• Page 5, Line 17 ff.: "To derive a lower estimate of uncertainty, we assumed

0.3 m uncertainty in the length of the rope at 16 m (resulting from knots tied into the rope) and neglected streching of the rope. This translates to (38.0+/-0.7) m. Further uncertainty is introduced by the image stitching and deskewing routines. The software estimates the internal and external camera orientation from the image data alone. Hence, the quality of the results strongly depends on the camera positions, image overlap and the object shape (Agisoft2016). In comparable applications, related errors in the millimeter and low centimeter range were found (e.g., Thoeni 2014, Robleda

2015). In our case they cannot be quantified and were assumed to be negligible."

**Referee comment**

P7, L1: What is the significance of the "large bedrock inclination"? Is this related to one of the components of the uncertainty, namely losing track of coherent phase?

Otherwise, this whole sentence seems to imply that there was in fact a component of uncertainty other than the two you discussed in section 2.3 but you got away with considering only the two by chance. Please clarify.

Keeping track of a coherent phase can be more difficult over an inclined bed.

Although most regions over NIF feature an almost planar bed (except over the crater rim) based on the referee's comment we feel it is necessary to explicitly refer to an additional effect: In regions with a large bed slope, a full 3-dimensional migration is superior but requires a sophisticated survey setup. With a 2- dimensional conventional migration ice thickness uncertainty is ~16% if the bed is strongly inclined (Moran and others, 2000). We thank the referee for pointing this out and have added specific reference to the above fact in section 2.3 and also changed the wording regarding P7 L1.

**Changes to manuscript:**

• Page 5, Lines 3-5: "In addition, in case of a strong..."

•   Page 7, Lines 11-13: "Since neither NIF2 nor NIF3 feature large surface/bed inclination (migration issues) nor pronounced presence of meltwater (Figure 4)

the uncertainty in GPR ice thickness seems to be well represented by our previous considerations."

•   We also decided against using the word "bedrock" to refer to the subglacial substrate, which at NIF consists to a large degree of sand. Accordingly we have replaced "bedrock" with simply "bed".

**Referee comment**

P7, L14-16: I don't agree that the observed mismatch could be attributed to the com- bined uncertainty. My interpretation of this statement is that your analysis of the com- bined uncertainty is wrong, which would require you to revise section 2.3. I don't think that is the case. It seems as though the mismatch could be largely due to the spatial and possibly the temporal variability (?) of the bottom melting caused by fumarole activities, which are not well documented so you are not able to quantify it, and a potential uncertainty in the core length.

Based on the referee's comment we realize that a different term should have been used than "observed mismatch", since there is no actual mismatch because the difference between ice loss values based on the GPR-ice core comparison and ablation stake measurements is in fact within the estimated range of uncertainties.

Hence we agree with the referee that this is not an issue of uncertainty considerations here. In fact, what we intend to discuss is the systematic offset (although within uncertainty) to larger ice loss derived from the GPR-ice core comparison. In this context, basal melting and uncertainty in ice core length could contribute to this offset but we are unable to quantify them. What we have tried to say is that, in view of the uncertainties involved, we cannot go as far as interpreting this result as evidence for basal melting. We have modified the wording of the respective paragraph to clarify.

**Changes to manuscript:**

•   Page 7, Lines 24-27: "In the absence of GPR evidence for basal fumarole activity and lacking quantitative information on basal melting, it seems more likely to attribute the observed systematic difference in the two ice loss estimates to the uncertainties involved in GPR and ablation stake measurements, combined with spatial variability of ablation rate and, to a minor extent, a potential discrepancy in the ice core length."

**Referee comment**

P8, L29-30: The discrepancy between your finding and the interpretation of Thompson et al. is significant. This warrants further discussions, at least further explain what

Thompson et al.'s interpretation is and more details on how your result questions their interpretation.

We have now added additional text in the discussion to clarify on the significance of our findings with respect to the study by Thompson et al. (2002). We also decided to move the discussion of the large dust layer in the NIF3 core from Page 8 Lines 27-29

to this section, since it illustrates the point being made here.

**Changes to manuscript:**

•   Changed paragraph starting on page 9, line 27: "With respect to the two ice core drilling sites..."

**Technical corrections**

These are very helpful and we have integrated all of the suggested corrections in the revised manuscript if not noted otherwise.

P2, L28: The use of the word "employed" is awkward. Change to "GPR has also been used..."

P2, L32: Add "e.g.," to the references because these might not be the only studies that used GPR on tropical glaciers.

P2, L32-33: "to our knowledge the study presented here..." should be '"to our knowl- edge this is the first time a GPR was used at Kilimanjaro's NIF."

P3, L3-5: The sentence "Although not further discussed..." seems unnecessary if not discussed at all in this manuscript.

We feel it is appropriate to keep this sentence, since it refers to the main reason why we extended our GPR profiles to precisely this position at the vertical wall. We also come back to this in the Conclusions.

P3, L5-6: The sentence should be changed to "We estimate the total ice volume presently remaining at NIF by spatially extrapolating the GPR-derived ice thickness."

P3, L8: Change "while" to "and".

P3, L9-10: You've defined the abbreviation already so use "IRH".

P3, L14: Change "as well as" to "and".

P3, L18: Change "employed" to "used".

P3, L18: Change "Technical settings of the setups" to "Details of the technical settings".

P3, L23: Change "The spatial coverage that could be achieved was constrained by" to

"The spatial extent of the GPR survey was constrained by ".

P3, L24: Change "employ" to "use".

P3, L27: Change "800 MHz profiles were not found to provide" to "800 MHz profiles did not provide".

P4, L5: I think "Post-processing of GPR data" reads better as a subsection heading.

P4, L6: "We used the standard routines to process the GPR data including ..."

P4, L9-11: The use of "while" in the sentence "We employed ..." is not appropriate so the sentence should be divided, with the first sentence ending after "5 traces" and the second sentence starting with "For the electromagnetic ...".

P4, L20: "Major contributions to the uncertainty in depth..."

P4, L21: Change "connected to" to "related to".

P4, L25: Change "loosing" to "losing".

P4, L26-27: You don't need the parenthesis.

P4, L29: Delete "relative difference".

P5, L8-9: Change "A 200 MHz CO-profile running towards the vertical wall extends to about one meter distance from the cliff" to "The 200 MHz CO-profile running towards the ice cliff ends within one meter from the cliff".

P5, L9: Change "The cliff height of the wall" to "The height of the ice cliff".

P5, L16: "In order to derive distributed ice thickness" to "To derive the ice-thickness distribution over the NIF", and remove the later "over the NIF".

P5, L16-17: Change "the previously developed approach by Fischer (2009), in interpolating" to "the approached previously developed by Fischer (2009), first interpolating".

P5, L21: "very high resolution" is subjective so remove "very".

P5, L22: No hyphen is needed for surface altitude.

P5, L33: Change "We derived an estimate" to "We estimated".

P6, L3: Change "In order to estimate the expected loss on surface area" to "To estimate the surface area lost".

P6, L14: Change "comprises" to "includes".

P6, L18: Change "reflectors from internal layers" to "internal reflectors".

P6, L19: Remove "very".

P6, L28: You don't need parentheses around the description of locations.

P6, L30: Delete ", however".

P7, L4: Remove "value".

P7, L13: "more or less" is ambiguous so remove.

P7, L17: Change "The interpolation of ice thickness" to "The interpolated ice thickness distribution".

P7, L28: Change "Considering additionally" to "In addition, considering".

P7, L28-29: Change "regard the values derived from this method with caution only" to

"interpret the ice thickness derived from this method with caution."

P8, L27: Change "large layer" to "thick layer".

P8, L29: Change "interpret" to "interpreted".

P8, L29: Remove "in depth".

P8, L30-32: It isn't totally clear whether "these findings" refer to your findings or those of Thompson et al. (I assume the former). Rewrite to clarify this.

P8, L30: Change "it seems worth" to "it is".

P9, L7: Change "near-bedrock ice parts" to "ice just above the bedrock".

P9, L28-29: Briefly explain why this finding is relevant for new ice core drilling and energy and mass balance modeling.

We have modified the sentence and added an additional reference.

P9, L31: Change "estimation" to "estimate".

P10, L2: Change "can be" to "were".

This is something you could sort out with TC's but I think figures are a little too small in general. Please pay particular attention to the size of texts embedded in each figures and make sure they are legible without blowing up on a computer screen. Labels of site and profile names in Figure 1, and legends in Figures 5 and 7 are particularly difficult to read.

We have taken care of the suggested changes and also generally tried to improve the readability of the figures by increasing font size etc.

Figures 1, 2 and 9: Label the top and bottom rows as (a) and (b), respectively, and refer to them accordingly in captions

---

## Editor Decision (ED1)

Comments from the Editor

The authors responded to reviewers adequately. However, this manuscript should be improved further to be accepted by the Cryosphere.

**Major scientific issues**

1. The main conclusion of this paper is the presence of uninterrupted, spatially coherent layering, but the presented evidence is weak.

1.1 Present much longer radar data. Now the authors show only 160-m-long profile (Fig. 2) and argue that the layering is well preserved in all profiles (it is said "all profiles" and later "nearly all profiles" or such, please be consistent). Apparently, the presented evidence is inadequate to support the claim. It is hard to see whether the radar reflectors are really continuous or not in Figure 4. Figure 9 can be more meaningful if more extensive radar data are presented.

1.2 Revise Figure 2b using multiple color (not gray scale) so that the layering structure can be more clearly seen.

1.3 Explain why 100-MHz radar data show less uninterrupted layering than 200 MHz. In general, lower frequency (longer wavelength) radar show more continuous layering. Why does this frequency difference occur, and why can you argue uninterrupted layering despite of limited features imaged by 100 MHz radar? (or why do you trust 200 MHz data more than 100 MHz data)

1.4 Abundant presence of meltwater found in shallow cores (P10L7; by the way how shallow are they?) infers the presence of isolated scatterers (percolated waterbodies into the deeper ice) and possible disturbance of the ice stratigraphy. With this shallow core evidence and inadequate presentation of the radar data, I cannot immediately support author's argument on the uninterrupted layering.

2. Ice thickness is estimated towards the western side of the NIF, where no radar data were collected (Fig. 7). However, ice thickness in that region is not at all data supported, and this affects the estimate of ice volume (Grid method). The authors discussed uncertainties in ice thickness, but such discussion can be valid within the area where data are present (central flat area). The sudden increase in the slope may be associated with the elevated bed near the boundary of the flat and steep areas (I.e. dam-up of the ice).

**Major presentation/structure issues**

1. Stake height changes are presented in Figure 5 and constitutes a major part of discussion in Section 2.5. However, it is not mentioned at all in the methods and suddenly appear in the results section. Please mention stake methods (i.e. locations of the stake, measurement periods etc) and AWS in Section 2.1.

2. The surface topography is shown only in Figure 6 but the authors say "flat central basin" from the beginning of the paper. Please re-arrange the figures so that the satellite image and surface topography are presented in Figure 1 to give the full topographic framework. Both of them are not author's original work so it can be presented as background knowledge.

**Minor points**

1. "internal" and "englacial" are used in an inter-changeable manner. Please use either of them consistently throughout the manuscript.
2. P1L6: add depth ranges of major englacial reflectors associated with dust layers.
3. P1L13f: Cite Figure 1 at the beginning part of Introduction (e.g. P1L17). Also, rearrange the figure so that Figure 6b (GeoEye-1 satellite imagery of Kilimanjaro) is presented as part of Figure 1 (see the major structure point #2 above).
4. P2L8: change "bed conditions" to "bed topography". Conditions sound like that the authors are primarily interested in whether the glacier has the cold bed or wet bed.
5. P2L9: remove "total"
6. P3L9: add "vertical" in front of discontinuities
7. P4L8: Please clearly mention that there is no/insignificant firn here, because firn affects the radio-wave propagation speed.
8. P4L27: how much of firn was found in the core? The authors simply said "negligible" but is it possible to shows an approximate fraction of firn and ice in the core?
9. P4L29: the authors interpreted the scattering near the surface exclusively caused by melt water. However, such scattering can occur with other causes, such as off-nadir crevasses or any structural features too (not in the plane of the radar profile).
10. P6L13: typo? "2011.46"? may be 2011.06??
11. P6L21-24: please revise. What do you mean by "all points"?
12. P7L2-3: cannot fully agree. Figure 1 shows patchy firn distributions (in the picture/image) and the vertical wall is in the blue ice area. The agreement at the wall does not validate the propagation speed and ice thickness measurement at the firn-covered area. Cross-over checks do not validate the propagation speed (as the same speed is used for both frequencies).
13. P8L22: revise to "with the presence of larger scattering near the surface" (it is not necessarily meltwater)
14. P8L26-28: The current flat surface does not imply the past flat surface (especially in this case where the ice is shrinking rapidly). Variable layer thickness can be caused by strain in the past. Also, ablation can happen from the surface or bottom but not inside of the ice body.
15. P8L29: please present the data. I cannot see any radar data supporting such localized layer convergence in the manuscript. Or do you refer gradual layer thickness change presented in Fig. 4?
16. Table 1: are samples for 200 MHz CMP measurements correct? Figure 3 looks like that there are more samples than 5.5 nsec/sample (= 100 nsec/18 samples). If it is not a typo and the sampling rate is so low, the data are not fully useful to determine the radio-wave propagation speed. Also, clarify "samples"; I understand that it is the number of samples within a time window (vertical range). Is it correct?
17. Table 3: does "relative depth" show the depth relative to the local ice thickness? Please clarify. And why are relative depths (in addition to the absolute depths) important for this context?
18. Figure 1: fill the area of tabular cliff with half-transparent color (or hatch). It is not easy to find out tabular cliff areas only using the outlines currently presented in this figure.
19. Figure 1: is it possible to add surface elevation contours to Figure 1? "the central flat area" is mentioned in Sections 1 and 2, but data supporting these sentences appear only in Figure 6. In general, the surface topography (and tabular cliffs) should be explained early in the manuscript, probably using a single paragraph in Section 1 (between "…. Kilimanjaro's glaciers to climate

variability." and "This especially …:" (P2L10). Also, include the AWS location in Figure 1 (it is referred several times in the text but its location is not shown).

20. Figure 4: The two core sites NIF2 and NIF3 are shown at the end of the profile. Please include radar data beyond these points so that radar data in the both sides of the core sites are presented.

21. Fig. 5's caption line 4: change "thick horizontal blue lines" to "thick horizontal blue markers", "bars" or such (confusing with the blue curves in the lower panel).

---

## Author Response (AR2)

**"Ground-penetrating radar reveals ice thickness and undisturbed englacial layers at**

**Kilimanjaro's Northern Ice Field" by Pascal Bohleber et al.**

- Response to Comments from the Editor -

***General Remarks:*** *All line numbers in "Changes to manuscript" refer to the revised version.*

*To differentiate from earlier changes made during the peer-review, new changes in the*

*corresponding pdf of the revised manuscript are now highlighted in blue.*

*Author's responses to the editor's comments are in blue.*

*All new references used in this text here can be found in the revised manuscript.*

**Comments from the Editor**

The authors responded to reviewers adequately. However, this manuscript should be improved further to be accepted by the Cryosphere.

We appreciate the editor's effort to help us to further improve the manuscript at this late stage of the process. We provide a response below and present the improved manuscript.

In doing so we believe we have further improved the scientific quality of this work and hope for a timely completion of the peer-review/editor review process.

**Major scientific issues**

1. The main conclusion of this paper is the presence of uninterrupted, spatially coherent layering, but the presented evidence is weak.

Demonstrating the existence of some spatially coherent layering is only one of the conclusions of the paper, which has also yielded the first map of ice thickness and permits volumetric estimation. This paper also provides a stratigraphic context for ice samples obtained during two prior expeditions, and which have yielded the first accurate $^{14}$C dates of Kilimanjaro ice.

We made an attempt to provide additional supporting evidence for our conclusion regarding layer coherence by including i) additional visual evidence of layering at the wall stratigraphy (Figure 8) and ii) showing all 200 MHz profiles in a new supplementary

Figure (also meeting the editor's request in 1.1 below).

We believe the evidence we provide (see the now included full set of radargrams in the supplementary figure) is strongly supporting coherent layering. This finding is not solely based on the GPR investigation but clearly supported by visual evidence from wall stratigraphy all around the NIF (revised Figure 8). Following the two peer reviews, in the revised manuscript we took additional care not to overstate our point regarding the layer coherence. For instance we specifically state that, as far as the GPR layers are concerned, we are referring to roughly the upper 30 m only and discuss limitations to the visibility of

GPR layers by noise from near-surface meltwater.

We made an attempt to provide additional supporting evidence for our conclusion regarding layer coherence by including i) additional visual evidence of layering at the wall stratigraphy (Figure 8) and ii) showing all 200 MHz profiles in a new supplementary

Figure (also meeting the editor's request in 1.1 below).

1.1 Present much longer radar data. Now the authors show only 160-m-long profile (Fig. 2)

and argue that the layering is well preserved in all profiles (it is said "all profiles" and later

"nearly all profiles" or such, please be consistent). Apparently, the presented evidence is inadequate to support the claim. It is hard to see whether the radar reflectors are really continuous or not in Figure 4. Figure 9 can be more meaningful if more extensive radar data are presented.

The purpose of Figure 2 is to provide a characteristic example of 100 and 200 MHz processed GPR profiles over the same horizontal distance. We intentionally restricted the horizontal distance to 160 m for better visibility of characteristic features such as noise by near-surface meltwater. We show an additional 150 m of 200 MHz profiles in Figure 4.

Regarding internal layering we refer to the 200 MHz profiles only, and argue that coherent internal layering is generally detected in all profiles. Attempting to quantify the extent to which the internal layers can be traced throughout the profiles we state on page 9, line 12-

14 that IRH4 is the deepest reflector that can be traced in almost all profiles. This is accurate, since it was not possible to trace IRH4 unambiguously over two short intervals, towards the eastern end of the plateau area and above the rise of the crater rim towards the west (this corresponds to the data gap in Figure 9 b) vs. a)).

In order to adequately address the request for more data we have made a new Figure that should be added to the paper as a supplement. It shows the entire 200 MHz profiles collected, divided into individual segments to aid visual perception. In our view the data clearly shows the major reflectors that extend throughout all profiles and which we associate with dust bands .

We also feel that it is necessary to point out that, compared to the typical standard in GPR profiles obtained over the interior of the polar ice sheets, not the same degree of clarity of IRH can be expected at warmer small scale mountain glaciers, and in particular at Kilimanjaro's NIF. This becomes especially evident with respect to the different ice formation process and the occasional presence of near-surface meltwater. In this context we use the term "uninterrupted" as the opposite of deformed, macroscopically disturbed layering. We have added text to clarify this more.

We would also like to point out that, with respect also to the 100 MHz profiles, we have already uploaded the entire dataset of ice thickness estimation based on the GPR bed reflection to the Pangaea repository (including both TWT and depth).

**Changes to manuscript:**
- Added a new Figure as supplementary material to show all 200 MHz processed GPR profiles
- Page 9, Line 19: "(as opposed to deformed, macroscopically disturbed layers)"

1.2 Revise Figure 2b using multiple color (not gray scale) so that the layering structure can be more clearly seen.

We have tried different color schemes and do not believe any scheme provides better visibility of layers . We have left the gray scale for Figure 2 but chose a color scale for the Supplementary Figure, thus providing both options for the reader.

1.3 Explain why 100-MHz radar data show less uninterrupted layering than 200 MHz. In general, lower frequency (longer wavelength) radar show more continuous layering. Why does this frequency difference occur, and why can you argue uninterrupted layering despite of limited features imaged by 100 MHz radar? (or why do you trust 200 MHz data more than 100 MHz data)

The answer to this is that while lower frequencies can penetrate deeper into the glacier, higher frequencies such as 200 MHz have better vertical (and horizontal) resolution.  Thus,

200 MHz provides the better image of the dielectric contrast produced by thin (dust)

layers. As an example for NIF, Thompson et al. (2002) report the most distinct visible dust layer to be 3 cm. This still results in a distinct reflector at the vertical resolution of 42 cm at

200 MHz. However, the 100 MHz profiles (at 84 cm vertical resolution) do not reveal a clear individual reflector anymore. We have added text to briefly explain this interpretation. Worth mentioning along these lines, other studies typically also chose to use frequencies of 250 MHz for investigating internal layers at small scale glaciers (e.g.

Eisen et al. 2003, Konrad et al. 2011).

In general, the editors question opens a wide field, best to be answered by multiple frequency surveys and measurement of dielectric properties at an ice core in high resolution. This is clearly off the focus and out of the possibilities of this study, but we will take that as suggestion for our next alpine coring.

**Changes to manuscript:**

• Page 6, Lines 32-33: "(due to the coarser vertical resolution at lower frequency)"

1.4 Abundant presence of meltwater found in shallow cores (P10L7; by the way how shallow are they?) infers the presence of isolated scatterers (percolated waterbodies into the deeper ice) and possible disturbance of the ice stratigraphy. With this shallow core evidence and inadequate presentation of the radar data, I cannot immediately support author's argument on the uninterrupted layering.

As stated on page 10, line 10, our shallow drillings reached only to typically about 0.6 m depth. Drilling deeper was severely hampered by water filling the holes. We discuss on page 10, section 3.3 that the presence of meltwater has been observed intermittently over the past years at NIF. This means at other times the glacier appears entirely frozen. At the time of our survey, meltwater was being produced at some locations and the GPR profiles show this accordingly (we agree that meltwater produces isolated scatterers and hence noise in our GPR profiles). The added Figure as a supplement shows the full extent of this effect, both laterally and in depth.

Regarding the effect of meltwater on internal layers, however, we believe that our conclusion regarding layer coherency remains valid, although meltwater-introduced noise near-surface can make the detection of IRH at depth more difficult (page 8, lines 22 ff.).

Notably we are already pointing out the relevance of the detected meltwater presence with respect to ice core records, i.e. especially concerning stable water isotopes that are known to be easily disturbed by meltwater.

2. Ice thickness is estimated towards the western side of the NIF, where no radar data were collected (Fig. 7). However, ice thickness in that region is not at all data supported, and this affects the estimate of ice volume (Grid method). The authors discussed uncertainties in ice thickness, but such discussion can be valid within the area where data are present (central flat area). The sudden increase in the slope may be associated with the elevated bed near the boundary of the flat and steep areas (I.e. dam-up of the ice).

As we discuss in section 2.1, it was not possible to walk everywhere with the GPR antennas due to rough surface terrain. We attempted to achieve the best possible coverage with our profiles, and our 100 MHz profiles extent over large portions of NIF, not just the central flat area. Nonetheless we are aware that the coverage is incomplete and the need for interpolation arises. This is in fact why we combine the GPR data with the DEM to interpolate ice thickness and estimate ice volume, because this means that additional constraint for interpolation was provided by the DEM.

**Major presentation/structure issues**

1. Stake height changes are presented in Figure 5 and constitutes a major part of discussion in Section 2.5. However, it is not mentioned at all in the methods and suddenly appear in the results section. Please mention stake methods (i.e. locations of the stake, measurement periods etc) and AWS in Section 2.1.

We thank the editor for pointing this out. Since Figure 5 was added somewhat late to the manuscript and is mostly based on published data we had not added details on the method.

Section 2.1 is exclusively dealing with the GPR survey setup, hence we decided to add details on the stake measurements at the respective first mentioning in the Introduction as well as by extending the caption of Figure 5. Locations of the stakes and measurement periods are all summarized in Figure 5 and we have added the location of the AWS in

Figure 5 and also in Figure 1 b).

**Changes to manuscript:**

•  Page 2, Lines 5-8: "comprehensive automatic weather stations (AWS) and network of mass balance stakes…"

•  Added AWS location to Figure 1b) and Figure 5

•  Page 7, Line 17-18: "the cumulative surface height change measured by two ultrasonic sensors at the AWS, close to NIF2, is -4.24 m."

•  Caption Figure 5: "Ice surface elevation change at NIF derived from ablation stakes with at least two consecutive measurements (increasing from n=1 to n=19 stakes, in

2000 and 2015, respectively). The AWS and spatial coverage of stakes at NIF are shown next to the legend in the upper left (black and red triangles, respectively). In the top plot, grey box plots represent the distribution or change in ice height (median, quartiles) at vertical or near-vertical stakes (< 30 tip; height measured along stake). Thick horizontal blue lines show the mean height change, or when only 1 measurement (i.e., 2001-2004)."

2. The surface topography is shown only in Figure 6 but the authors say "flat central basin"

from the beginning of the paper. Please re-arrange the figures so that the satellite image and surface topography are presented in Figure 1 to give the full topographic framework.

Both of them are not author's original work so it can be presented as background knowledge.

Although we see the logic behind this comment, we doubt rearranging the Figures would be beneficial to the reader, since we have deliberately chosen to show the individual details of the Figures for the following reasons: The reason for not showing contour lines of topography in Figure 1 is that having a second set of lines makes it more difficult to recognize the GPR profile lines – which is the more-important element of the paper. The reason why the satellite image is in Figure 6 is the discussion of the crater rim, and we have enlarged the image to become a separate part of Figure 6 following one of the referee's comments. The surface topography is shown again together with the GPR ice thickness in Figure 6 because these are both input datasets for the interpolation of ice thickness in Figure 7.

**Minor points**

1. "internal" and "englacial" are used in an inter-changeable manner. Please use either of them consistently throughout the manuscript.

Wherever interchangeable, we have changed "englacial" to "internal" throughout the manuscript. However, we would like to keep the original title of the manuscript.

2. P1L6: add depth ranges of major englacial reflectors associated with dust layers.

We have already added in the revised manuscript making references to the depth ranges in the abstract: Page 1, Line 8 "at least for the upper 30 m"

3. P1L13f: Cite Figure 1 at the beginning part of Introduction (e.g. P1L17). Also, rearrange the figure so that Figure 6b (GeoEye-1 satellite imagery of Kilimanjaro) is presented as part of Figure 1 (see the major structure point #2 above).

We have added citing Figure 1 on Page 2, Line 11. For the reasons stated above (major point #2) we are not rearranging the Figures.

4. P2L8: change "bed conditions" to "bed topography". Conditions sound like that the authors are primarily interested in whether the glacier has the cold bed or wet bed.

We wanted to also point to the fact that little is known about the bed conditions, although we are of course mainly interested in the topography. We have changed this accordingly, now saying on Page 2, Line 9 "bed conditions and topography".

5. P2L9: remove "total"

Done.

6. P3L9: add "vertical" in front of discontinuities

Done.

7. P4L8: Please clearly mention that there is no/insignificant firn here, because firn affects the radio-wave propagation speed.

Done. Page 4, Line 11-12: "Because of the insignificant amount of firn at NIF,…"

8. P4L27: how much of firn was found in the core? The authors simply said "negligible" but is it possible to shows an approximate fraction of firn and ice in the core?

Judging from Figure S1 of Thompson et al. (2002) and assuming firn was defined by its density, the firn part in the ice core is less than 10 cm deep. It is also worth mentioning that, if firn is defined as snow which has endured an ablation season, became more dense, and was buried by subsequent accumulation, there is none this century at NIF. Snow on the

NIF either sublimates, or melts and then either runs off and/or down – or the meltwater refreezes at the surface as superimposed ice, see Hardy (2011) for more details on this.

9. P4L29: the authors interpreted the scattering near the surface exclusively caused by melt water. However, such scattering can occur with other causes, such as off-nadir crevasses or any structural features too (not in the plane of the radar profile).

Based on our experience with the drilling attempts in the field, melt water seems the most likely cause. This is also due to the fact that, with one exception, we did not observe any crevasses, cracks etc.

10. P6L13: typo? "2011.46"? may be 2011.06??

No, this is a decimal date as it is used in the original publication by Cullen et al. (2013).

11. P6L21-24: please revise. What do you mean by "all points"?

We changed "all points" to "all data points" to make this more clear.

12. P7L2-3: cannot fully agree. Figure 1 shows patchy firn distributions (in the picture/image) and the vertical wall is in the blue ice area. The agreement at the wall does not validate the propagation speed and ice thickness measurement at the firn-covered area. Cross-over checks do not validate the propagation speed (as the same speed is used for both frequencies).

1. Please note that the satellite image was recorded at a different date than the GPR survey.

More importantly, however, the amount of firn is generally negligible, as argued above.

During the GPR survey, the surface conditions at the wall were highly similar to the interior surface (Figure 8, a)). Accordingly, we are convinced that, as compared to other glaciers not being of the tabular structure, the wall does in fact provide a unique opportunity to check ice thickness sounding and have made an attempt to take advantage of this.

2. We mainly used the cross-over checks to demonstrate consistency in bed detection using

100 and 200 MHz. We have clarified this. Page 7, Line 6: "…values for ice thickness are consistent within their uncertainty"

13. P8L22: revise to "with the presence of larger scattering near the surface" (it is not necessarily meltwater)

Considering our reply above considering meltwater being the most likely cause of the near- surface scattering, we have changed the text to: Page 8, Line 24: "…coincide with a large amount of near-surface scattering, presumably due to the presence of near-surface meltwater."

14. P8L26-28: The current flat surface does not imply the past flat surface (especially in this case where the ice is shrinking rapidly). Variable layer thickness can be caused by strain in the past. Also, ablation can happen from the surface or bottom but not inside of the ice body.

We appreciate the input but are not sure if there is actually a disagreement here. We were
not trying to say that ablation happens inside the ice body (hard to imagine how this would
work) but in fact, our point is that we believe the observed features are related to ablation
as opposed to rheology.

15. P8L29: please present the data. I cannot see any radar data supporting such localized
layer convergence in the manuscript. Or do you refer gradual layer thickness change
presented in Fig. 4?
We are not referring to the gradual layer thickness change but mean actual convergence of
two layers into one layer, which can only be observed close to the crater rim. As requested
we are now showing the respective data in our supplementary Figure (Profile D). No layer
convergence is seen towards the ice cliff or in the interior.

16. Table 1: are samples for 200 MHz CMP measurements correct? Figure 3 looks like that
there are more samples than 5.5 nsec/sample (= 100 nsec/18 samples). If it is not a typo
and the sampling rate is so low, the data are not fully useful to determine the radio-wave
propagation speed. Also, clarify "samples"; I understand that it is the number of samples
within a time window (vertical range). Is it correct?
In case of the CMP, the number of samples refers to the number of shots of the CMP, e.g. the
number of times the antennas were repositioned. Thank you for pointing this out, we have
clarified this in the Table caption.

17. Table 3: does "relative depth" show the depth relative to the local ice thickness? Please
clarify. And why are relative depths (in addition to the absolute depths) important for this
context?
Yes, relative depth means relative to local ice thickness (which is always at 100%). This
change was made in the revised manuscript specifically to meet a reviewer's comment,
suggesting this for aiding the comparison of IRH depths.

18. Figure 1: fill the area of tabular cliff with half-transparent color (or hatch). It is not easy
to find out tabular cliff areas only using the outlines currently presented in this figure.

We do not believe the reader would benefit from adding any more detail to Figure 1. As
said earlier, the main purpose of Figure 1 is to show the locations of the GPR profiles.
However, we made an attempt to address this comment by adding to Figure 8 more
pictures that clearly show the cliff locations on NIF.
19. Figure 1: is it possible to add surface elevation contours to Figure 1? "the central flat
area" is mentioned in Sections 1 and 2, but data supporting these sentences appear only in
Figure 6. In general, the surface topography (and tabular cliffs) should be explained early
in the manuscript, probably using a single paragraph in Section 1  (between "….
Kilimanjaro's glaciers to climate variability." and "This especially …:" (P2L10). Also, include
the AWS location in Figure 1 (it is referred several times in the text but its location is not
shown).
See the comment made above regarding visibility of the GPR profiles, we believe it is better
to leave out contour lines in Figure 1. However, we have added the position of the AWS to
Figure 1 b) and also Figure 5. We have also added text to the introduction explaining the
surface topography earlier in the text: Page 3, Line 1-2: "Typical for the tabular glaciers on
Kilimanjaro's summit (cf. slope glaciers) the NIF topography is characterized by a central
flat plateau area and near-vertical ice margins (Kaser et al., 2004; Cullen et al., 2006;
Hardy, 2011)."
20. Figure 4: The two core sites NIF2 and NIF3 are shown at the end of the profile. Please
include radar data beyond these points so that radar data in the both sides of the core sites
are presented.
We have included this request in the new supplementary Figure showing all 200 MHz
profiles. We have indicated the positions of NIF2, NIF3 and the intersection, analog to what
is shown in Figure 4.
21. Fig. 5's caption line 4: change "thick horizontal blue lines" to "thick horizontal blue
markers", "bars" or such (confusing with the blue curves in the lower panel).
Done.

---

## Author Response (AR3)

Dear Editor Kenny Matsuoka,

Thank you for getting back to us quickly after returning from your fieldwork. We are happy to hear that you find our revised manuscript improved and ready to be accepted by TC. We have taken care of the technical corrections and are uploading a final revised version of the manuscript. All new changes to the text are now marked in green and we have included a short reply to your comments below.

Thank you again for your help in further improving the manuscript.

Kind regards,

Pascal Bohleber, on behalf of all co-authors

**Response to comments by the editor/ technical corrections**

Dear authors,

Thank you for submitting the revised manuscript timely and for being patient while I deployed to a field camp in Antarctica. The new supplement figure (all radargrams) constitutes strong evidence of author's argument, and editing in this stage clarified many issues. So I am happy to accept this manuscript with technical corrections.

- In conclusion, the authors argue that the ice stratigraphy is preserved at least in the top 30 m, which is supported by radar data. Separately, the authors mention that chemical and isotope records may be disturbed in the top 10 m at the very end of discussion (Section 3.3). I judge that both statements are valid but request a more synthesized statement in the conclusion, such as "Macroscopic coherence of the radar data infers uninterrupted ice in the top 30 m, but abundant melt water could potentially collapse chemical and isotope records in the top 10 m."

We have changed the wording of the respective sentence and now provide the requested synthesis statement in the conclusion.

**Changes to manuscript:**

- Page 10, Line 21-23: "For the central former drilling area, the radar profiles reveal macroscopic coherent, uninterrupted ice layering for at least the upper 30 m, and demonstrate abundant melt water in the top 10 m. The latter finding suggests that the upper part of future chemical and isotopic ice core records could potentially be corrupted by meltwater."

- In table 2, show area in the unit of 10^6 m^2 only to justifiable significant digits.

Changed accordingly.

- Regarding minor points #14 and #15, the main source of confusion is, I believe, that the manuscript does not distinguish (1) radar reflector and (2) ice layer that is bounded by two adjacent reflectors. Convergence of two radar reflectors make a single narrowing ice layer. Please revise the relevant text.

We have revised the text to make this more clear.

**Changes to manuscript:**

- Page 8, Lines 31-32: "The GPR profiles towards the western end are the only case in which adjacent IRH (representing boundaries to a layer of ice) are found merging together."

- Be more specific in the data availability; I.e. "ice thickness along all radar profiles are available at…"

Changed accordingly.

- Clarify non-lat/lon coordinates in Figures 1, 6, 7, and 9 (is it UTM? If so, specify the zone).

Changed accordingly. It is UTM 37M.

- In the supplement figure, does a gray curve show the outline of the cliff? Clarify it in the caption.

Changed accordingly. Yes, the grey curve shows the outline of the cliff.

Thank you for submitting your work in the journal The Cryosphere

Kenny Matsuoka
TC/TCD Editor